YITP-SB-17-8

# Crossing Kernels for Boundary and Crosscap CFTs

**Matthijs Hogervorst**

C.N. Yang Institute for Theoretical Physics, Stony Brook University, USA

**Abstract**

This paper investigates $d$-dimensional CFTs in the presence of a codimension-one boundary and CFTs defined on real projective space $\mathbb{RP}^d$. Our analysis expands on the alpha space method recently proposed for one-dimensional CFTs in Ref. [1]. In this work we establish integral representations for scalar two-point functions in boundary and crosscap CFTs using plane-wave-normalizable eigenfunctions of different conformal Casimir operators. CFT consistency conditions imply integral equations for the spectral densities appearing in these decompositions, and we study the relevant integral kernels in detail. As a corollary, we find that both the boundary and crosscap kernels can be identified with special limits of the $d = 1$ crossing kernel.

March 2017

# 1  Introduction

It has been known for a long time that conformally invariant quantum field theories are strongly constrained by symmetry arguments and unitarity. The conformal bootstrap program, envisioned in the 1970s [2–9], aims to maximally exploit these constraints. In two dimensions much progress has been made through purely algebraic methods, starting from Ref. [10]. In recent years the bootstrap program for CFTs in higher spacetime dimensions has been revived [11], leading to a variety of numerical and analytical results, see e.g. [12–32].

Most bootstrap results so far have been obtained by studying CFT correlators in position space. For numerical purposes, studying position-space bootstrap equations — using the language of linear functionals — has led to powerful algorithms [33] which allow for the determination of CFT spectra with high numerical precision (see e.g. [29, 32] for state-of-the-art results). Still, it seems possible that a deeper analytical understanding of CFTs could be obtained using a different mathematical framework. One example of such a framework is Mellin space [34–42], which has been instrumental both in understanding holographic aspects of large-$N$ CFTs as well as in finding exact results for Wilson-Fisher and $O(N)$ fixed points. A second example is furnished by the Froissart-Gribov formula from Ref. [43] — or see [44] for a similar setup in one dimension. More tentatively, it is now known that conformal blocks can be mapped to the Calogero-Sutherland model [45, 46], providing at least a formal relation between non-integrable CFTs and integrable Hamiltonians.

In recent work with B.C. van Rees, we developed a different approach to analyze CFT correlators, known as *alpha space* [1]. In the paper in question, four-point functions in one-dimensional CFTs were examined using the Sturm-Liouville theory of the conformal Casimir operator. Roughly speaking, alpha space denotes an integral transform which maps a four-point function $F(z)$ in position space to a complex density $\widehat{F}(\alpha)$, where $\alpha$ runs over a complete basis of eigenfunctions of the $1d$ Casimir. By construction, the poles and residues of $\widehat{F}(\alpha)$ are in one-to-one correspondence with (experimentally relevant) CFT data: the scaling dimensions and OPE coefficients appearing in the conformal block decomposition of $F(z)$. Finally, it can be shown that crossing symmetry forces $\widehat{F}(\alpha)$ to satisfy a specific integral equation. The relevant integration kernel, which we denote as $K(\alpha, \beta)$, plays a crucial role: it is a new, universal object in CFT which completely encodes crossing symmetry.[1]

Although one-dimensional bootstrap equations are phenomenologically interesting, it is natural to try and extend the alpha space formalism to a more general setting. An ambitious goal would be to construct the crossing kernel for general four-point functions in $d$ dimensions. This appears to be a rather complicated exercise, closely related to computing the $6j$ or Racah-Wigner symbols of the conformal algebra $\mathfrak{so}(d + 1, 1)$ (see e.g. [48]). In the present work we focus on a different problem, extending the alpha space formalism to the case of two-point correlators of $d$-dimensional CFTs in the presence of non-trivial backgrounds.

In the first part of this paper, we consider boundary CFTs (BCFTs). These are CFTs defined on half of Euclidean space, having a conformally invariant boundary of codimension one. Conformal symmetry puts stringent constraints on the space of possible boundary conditions; especially in $d = 2$, these constraints have been explored in detail [49–61].[2] Kinematical constraints on $d$-dimensional BCFTs have been worked out in detail in Refs. [66–68], paving the way for more recent

---

[1]In a more abstract setting, the existence of such a crossing kernel was already mentioned in Ref. [47].
[2]See also Refs. [62–65] for more pedagogical material.

bootstrap analyses [69, 70]. See also Refs. [71, 72] for a discussion of more general (supersymmetric) defects in the context of the conformal bootstrap.

In the second part we turn our attention to CFTs on $d$-dimensional real projective space $\mathbb{RP}^d$. In $d = 2$ this manifold is referred to as the *crosscap*. By abuse of language, we will employ this term for the $d$-dimensional case as well. One-point functions on the crosscap background are generally non-vanishing, but the relevant VEVs are constrained in a way that is similar to boundary CFTs [73, 74]. Crosscap CFTs have received a significant amount of attention due to their role in the AdS/CFT correspondence [75–80]. Apart from any holographic applications, crosscap bootstrap equations on $\mathbb{RP}^d$ have recently been used to investigate specific CFTs in $d > 2$ dimensions [81, 82]: numerically in the case of the $3d$ Ising model, and analytically in the case of the Lee-Yang fixed point in $6 - \epsilon$ dimensions.

In this paper we do not aim to constrain specific boundary or crosscap CFTs. Our primary goal is to develop a better understanding of the alpha space method of [1], and to provide a detailed comparison between the $d = 1$ crossing symmetry kernel $K(\alpha, \beta)$ and the kernels that arise in the boundary and crosscap context in $d$ dimensions. We will only summarily comment on concrete bootstrap applications, focusing on more mathematical aspects of the alpha space formalism.

The outline of this paper is as follows. In Section 2, we provide a somewhat pedagogical introduction to the alpha space formalism using a toy example. We then review the Jacobi transform in Sec. 3. This is a classical integral transform which already appeared in the alpha space analysis of $d = 1$ CFTs. In Sec. 4 we analyze BCFT bootstrap equations for scalar two-point functions from the alpha space point of view. We will first develop integral representations for these correlators, and then use these representations to define a crossing kernel which will be the BCFT counterpart of the $d = 1$ kernel $K(\alpha, \beta)$. Section 5 develops the alpha space formalism for correlators on $\mathbb{RP}^d$. Sec. 6 is the most important part of this paper: there, we analyze the different crossing kernels. In Sec. 7 we discuss several directions for future work, and in Appendix A we compute several concrete alpha space transforms.

## 2  Warm-up: toy bootstrap in alpha space

For pedagogical purposes, we will work out an example of an alpha space transform in a toy example, including the computation of a crossing kernel. Our example is defined as follows. Consider a function $F(x)$ on the interval $[0, 1]$ which can be expanded as

$$F(x) = \sum_k c_k \, x^{\Delta_k} \tag{2.1}$$

and obeys the following functional equation:

$$F(x) = \left( \frac{x}{1 - x} \right)^{\Delta_\phi} F(1 - x) \,. \tag{2.2}$$

For now $\Delta_\phi$ is an arbitrary parameter. Eq. (2.2) is a generic example of a CFT crossing equation: for instance, one encounters (2.2) after restricting a $d$-dimensional bootstrap equation to the diagonal $z = \bar{z}$ [83]. In a realistic setting, $\Delta_\phi$ would play the role of the scaling dimension of an external primary operator. The decomposition (2.1) treats all states, primaries and descendants,

democratically. In actual CFT examples the basis functions will not be power laws $x^\Delta$ — rather, we will have to use certain special functions known as conformal blocks.

In the alpha space framework, one starts by thinking of the "blocks" $x^\Delta$ as solutions of a second-order ODE, namely

$$D_{\text{toy}} \cdot x^\Delta = C_d(\Delta)x^\Delta\,, \quad D_{\text{toy}} = (1 - 2h)x\partial_x + x^2\partial_x^2\,. \tag{2.3}$$

Here $C_d(\Delta) = \Delta(\Delta - d)$ is the Casimir eigenvalue for a scalar of dimension $\Delta$ in $d$ dimensions, and throughout this paper we use the shorthand notation

$$h \equiv d/2\,. \tag{2.4}$$

Our first goal is to decompose $F(x)$ in terms of a complete basis of eigenfunctions of $D_{\text{toy}}$. In order to do so, we notice that $D_{\text{toy}}$ is self-adjoint with respect to the inner product

$$\langle f, g\rangle_{\text{toy}} = \int_0^1 \frac{dx}{x^{2h+1}} \, \overline{f(x)}g(x)\,. \tag{2.5}$$

In order to find such a basis, we impose the boundary condition $f(1) = 1$. This singles out the following family of eigenfunctions of $D_{\text{toy}}$:

$$\chi_\alpha(x) = x^h \cosh(\alpha \ln x) = \frac{x^{h+\alpha} + x^{h-\alpha}}{2} \tag{2.6}$$

which are labeled by an *a priori* complex parameter $\alpha$. If $\Re(\alpha) \neq 0$ the above functions are not normalizable with respect to (2.5). However, we claim that the $\chi_\alpha$ with *imaginary* $\alpha$ form an orthogonal basis on $[0, 1]$. To check orthogonality, one simply computes the inner product

$$\langle \chi_{ia}, \chi_{ib}\rangle_{\text{toy}} = \int_0^1 \frac{dx}{2x} \left[\cos\big((a + b)\ln x\big) + \cos\big((a - b)\ln x\big)\right]$$
$$= \frac{\pi}{2}\left[\delta(a + b) + \delta(a - b)\right] \tag{2.7}$$

writing $\alpha = ia$ and $\beta = ib$ to make it clear that this expression holds for imaginary $\alpha, \beta$. From (2.7) one derives completeness, i.e. the fact that a test function $f(x)$ can be decomposed as

$$f(x) = \frac{1}{\pi} \int [d\alpha]\, \widehat{f}(\alpha)\chi_\alpha(x) \;\leftrightarrow\; \widehat{f}(\alpha) = \int_0^1 \frac{dx}{x^{2h+1}}\, f(x)\chi_\alpha(x)\,. \tag{2.8}$$

Throughout this paper we use the notation

$$\int [d\alpha] = \frac{1}{2\pi i} \int_{c-i\infty}^{c+i\infty} d\alpha \tag{2.9}$$

to denote Mellin contours, running along the imaginary axis. Such integrals are always to be understood in the Mellin-Barnes sense, meaning that when necessary the integration contour must be deformed to separate families of poles that go to the right from those that go to the left.

Eq. (2.8) is our first example of an alpha space transform, mapping an arbitrary position-space function $f(x)$ to a complex density $\widehat{f}(\alpha)$. This density is naturally defined for $\alpha$ on the imaginary axis, but we will often need to analytically continue to other regions of the complex

plane. From a mathematical point of view, Eq. (2.8) is well-defined for functions $f(x)$ that are square-normalizable with respect to (2.5). For more general position-space functions, the same integral transform still makes sense after deforming the integration contour on the alpha space side — see e.g. [1] where this is made precise. In passing, we notice that (2.8) is nothing but the Fourier-cosine transform, as can be seen by changing coordinates $x \to e^{-y}$.

Let us now apply the above transform to the correlator $F(x)$ appearing in our bootstrap problem. Following (2.8), we start by writing down the following integral representation for $F(x)$:

$$F(x) = \frac{1}{\pi} \int_{\mathcal{C}} [d\alpha] \, \widehat{F}(\alpha) \chi_\alpha(x) \tag{2.10}$$

and we will attempt to obtain constraints on the density $\widehat{F}(\alpha)$. The contour $\mathcal{C}$ needs to be specified: if $F(x)$ is normalizable we can take $\mathcal{C}$ to coincide with the imaginary axis, otherwise $\mathcal{C}$ needs to be deformed. We will first argue that Eq. (2.1) can be derived from (2.12). To see this, note that $\chi_\alpha(x)$ is even in $\alpha$, hence $F(x)$ can be rewritten as

$$F(x) = \frac{1}{\pi} \int_{\mathcal{C}} [d\alpha] \, \widehat{F}(\alpha) x^{h+\alpha} \,. \tag{2.11}$$

Now assume that $\widehat{F}(\alpha)$ decays sufficiently fast on the right half plane, such that the contour can be closed. Then by Cauchy's theorem, Eq. (2.11) can be recast as

$$F(x) = -\frac{1}{\pi} \sum_k \mathrm{Res} \, \widehat{F}(\alpha)\big|_{\alpha=\alpha_k} x^{h+\alpha_k} \tag{2.12}$$

with the sum running over all poles $\alpha_k$ inside $\mathcal{C}$, circled in the positive direction. We see that picking up the poles inside the contour $\mathcal{C}$ has yielded a decomposition of exactly the same form as (2.1); in order to match both sets of parameters, we must have

$$\Delta_k = h + \alpha_k \quad \text{and} \quad c_k = -\frac{1}{\pi} \mathrm{Res} \, \widehat{F}(\alpha)\big|_{\alpha=\alpha_k} \,. \tag{2.13}$$

This point explains why it was necessary to decompose $F(x)$ using eigenfunctions of $D_{\mathrm{toy}}$ — we could certainly have used any other complete set of functions, but doing so would not allow us to match the poles and residues of $\widehat{F}(\alpha)$ to CFT data.

At this point, we would like to constrain the complex density $\widehat{F}(\alpha)$. To do so, we insert the integral decomposition (2.12) into the crossing equation (2.2). This yields

$$\int [d\alpha] \, \widehat{F}(\alpha) \chi_\alpha(x) = \int [d\beta] \, \widehat{F}(\beta) \left( \frac{x}{1-x} \right)^{\Delta_\phi} \chi_\beta(1-x) \,. \tag{2.14}$$

To proceed, we remark that the functions $\chi_\alpha(x)$ form a complete basis on $[0,1]$. We can therefore write

$$\left( \frac{x}{1-x} \right)^{\Delta_\phi} \chi_\beta(1-x) = \frac{1}{\pi} \int [d\alpha] \, K_{\mathrm{toy}}(\alpha, \beta) \chi_\alpha(x) \tag{2.15}$$

for some kernel $K_{\mathrm{toy}}(\alpha, \beta)$, which can depend on $d$ and the label $\Delta_\phi$. Using (2.8), we find that it is given by the following expression:

$$K_{\mathrm{toy}}(\alpha, \beta) = \int_0^1 \frac{dx}{x^{2h+1}} \left( \frac{x}{1-x} \right)^{\Delta_\phi} \chi_\beta(1-x) \chi_\alpha(x) \,. \tag{2.16}$$

Eqs. (2.15) and (2.16) have a simple interpretation, namely that $K_{\text{toy}}(\alpha, \beta)$ defines a change-of-basis matrix which relates the cross-channel functions appearing on the LHS of (2.15) to the basis functions $\chi_\alpha(x)$. Finally, plugging (2.15) into (2.14), we conclude that $\widehat{F}(\alpha)$ satisfies the following integral equation:

$$\widehat{F}(\alpha) = \frac{1}{\pi} \int [d\beta]\, K_{\text{toy}}(\alpha, \beta) \widehat{F}(\beta)\,. \qquad (2.17)$$

The integral (2.16) is of course easy to compute, yielding

$$K_{\text{toy}}(\alpha, \beta) = k(\alpha, \beta) + k(-\alpha, \beta) + k(\alpha, -\beta) + k(-\alpha, -\beta) \qquad (2.18)$$

where

$$k(\alpha, \beta) = \frac{1}{4} \mathrm{B}(\alpha + \Delta_\phi - h, \beta - \Delta_\phi + h + 1)\,, \quad \mathrm{B}(x, y) := \frac{\Gamma(x)\Gamma(y)}{\Gamma(x + y)}\,. \qquad (2.19)$$

Eq. (2.16) converges only if $h < \Delta_\phi < h + 1$. Outside of this interval, the formula on the RHS of (2.19) provides an analytic continuation.

Taking a step back, we conclude that solving the toy bootstrap equation (2.2) is equivalent to finding a complex function $\widehat{F}(\alpha)$ with the following properties:

- $\widehat{F}(\alpha)$ is even in $\alpha$;

- $\widehat{F}(\alpha)$ is meromorphic, all its poles lie on the real axis and have real-valued residues;

- $\widehat{F}(\alpha)$ obeys the integral equation (2.17).

Requiring positivity of the coefficients $c_k$ appearing in (2.2) — as one commonly does in the conformal bootstrap — would additionally imply that the residues of $\widehat{F}(\alpha)$ are negative, rather than simply real-valued. This follows from (2.13).

Since this is a toy example we refrain from examining the kernel (2.18) in detail, nor will we try to construct solutions to (2.17). However, we stress that $K_{\text{toy}}(\alpha, \beta)$ is a relatively simple function of two complex variables. For one, making the dependence of $K_{\text{toy}}$ on the external label $\Delta_\phi$ explicit, the kernel obeys a "duality" relation

$$K_{\text{toy}}(\alpha, \beta | \Delta_\phi) = K_{\text{toy}}(\beta, \alpha | 2h + 1 - \Delta_\phi) \qquad (2.20)$$

which relates its two arguments. Second, in both of its arguments, $K_{\text{toy}}(\alpha, \beta)$ is a meromorphic function with integer-spaced poles. Moreover, the residues at these poles are polynomials. For instance, keeping $\beta$ fixed, the kernel has two series of poles in $\alpha$, at $\pm\alpha = \Delta_\phi - h + \mathbb{N}$, and the residues at these poles are given by

$$- \operatorname{Res} K_{\text{toy}}(\alpha, \beta)\big|_{\alpha = \Delta_\phi - h + n} = \frac{1}{4n!} \left[ (\Delta_\phi - h + \beta)_n + (\Delta_\phi - h - \beta)_n \right]\,. \qquad (2.21)$$

Here $(x)_n = \Gamma(x + n)/\Gamma(x)$ denotes the Pochhammer symbol.

# 3  Review of the Jacobi transform

In the rest of this paper, we will use alpha space technology to examine actual CFT bootstrap equations. Our computations will be similar to the toy computation from Sec. 2, the main difference being that the relevant correlation functions will have non-trivial conformal block decompositions, contrary to Eq. (2.1). Consequently, we will have to deal with more complicated integral transforms. It turns out that all of our examples can be treated systematically by introducing the *Jacobi transform*, which denotes a two-parameter family of integral transforms. Essentially, this is the Sturm-Liouville theory of the differential operator

$$\mathcal{D}_{p,q} = -\left[p + q + (1-q)x\right]x\partial_x + (1-x)x^2\partial_x^2 \tag{3.1}$$

on the interval $[0,1]$. In what follows we will refer to $\mathcal{D}_{p,q}$ as the Jacobi operator. For now $p, q$ are two fixed, arbitrary parameters; in the CFT settings we consider, they will be related to the spacetime dimension $d$ and the scaling dimensions of external operators. In the following section we will reproduce some results from the literature, without aiming for any level of mathematical rigor. For a more extensive discussion, see Refs. [84–88].

For the Sturm-Liouville analysis of $\mathcal{D}_{p,q}$ on $[0,1]$, we start by noticing that $\mathcal{D}_{p,q}$ is self-adjoint with respect to the inner product[3]

$$\langle f, g\rangle_{p,q} = \int_0^1 dx\, w_{p,q}(x)\,\overline{f(x)}g(x)\,, \quad w_{p,q}(x) = \frac{(1-x)^p}{x^{2+p+q}}\,. \tag{3.2}$$

Let us denote the Hilbert space of square-normalizable functions by $\mathfrak{H} = \mathfrak{H}_{p,q}$. We claim that a complete and orthogonal basis is given by the so-called Jacobi functions:

$$\vartheta_\alpha^{(p,q)}(x) = {}_2F_1\left(\begin{matrix} \frac{1}{2}(1+p+q)+\alpha,\ \frac{1}{2}(1+p+q)-\alpha \\ 1+p \end{matrix}; \frac{x-1}{x}\right) \tag{3.3}$$

with $\alpha \in i\mathbb{R}$, which are eigenfunctions of $\mathcal{D}_{p,q}$ with eigenvalue $\alpha^2 - \frac{1}{4}(1+p+q)$. The Jacobi functions will play a similar role as the basis functions $\chi_\alpha(x)$ did in the previous section. To prove orthogonality, we compute

$$\langle \vartheta_{ia}, \vartheta_{ib}\rangle_{p,q} = \frac{\mathcal{N}_{p,q}(ia)}{2}\left[\delta(a+b) + \delta(a-b)\right] \tag{3.4}$$

where

$$\mathcal{N}_{p,q}(\alpha) = \frac{|Q_{p,q}(\alpha)|^2}{2}\,, \quad Q_{p,q}(\alpha) = \frac{2\Gamma(1+p)\Gamma(-2\alpha)}{\Gamma\left(\frac{1}{2}(1+p+q)-\alpha\right)\Gamma\left(\frac{1}{2}(1+p-q)-\alpha\right)}\,. \tag{3.5}$$

Using (3.4), we find the following pair of integral transforms:

$$f(x) = \int \frac{[d\alpha]}{\mathcal{N}_{p,q}(\alpha)}\,\widehat{f}(\alpha)\vartheta_\alpha^{(p,q)}(x) \;\leftrightarrow\; \widehat{f}(\alpha) = \int_0^1 dx\, w_{p,q}(x)\, f(x)\vartheta_\alpha^{(p,q)}(x)\,. \tag{3.6}$$

Strictly speaking, the Jacobi transform $f(x) \mapsto \widehat{f}(\alpha)$ is a map from $\mathfrak{H}$ to the Hilbert space $\mathfrak{A}_{p,q}$ of functions $\phi(\alpha) = \phi(-\alpha)$ on the imaginary axis, normalizable with respect to the inner product

$$(\phi, \chi)_{p,q} = \int \frac{[d\alpha]}{\mathcal{N}_{p,q}(\alpha)}\,\overline{\phi(\alpha)}\chi(\alpha)\,. \tag{3.7}$$

---

[3]Our conventions differ from those in Ref. [1] as follows: we have $x_{\text{here}} = 1/(1+x_{\text{there}})$.

Moreover, it follows from (3.6) that this map is unitary (i.e. an isometry): for any two normalizable position-space functions $f, g$ we have

$$\langle f, g \rangle_{p,q} = \left( \widehat{f}, \widehat{g} \right)_{p,q}, \quad f, g \in \mathfrak{H}. \tag{3.8}$$

Using the same logic as the toy example, the Jacobi transform can be used to reproduce a conformal block decomposition. For $p = q = 0$ this was shown in [1]; here we will briefly review the argument. Consider a function $f(x)$ decomposed in terms of Jacobi functions, as in Eq. (3.6):

$$f(x) = \int_{\mathcal{C}} \frac{[d\alpha]}{\mathcal{N}_{p,q}(\alpha)} \, \widehat{f}(\alpha) \vartheta_{\alpha}^{(p,q)}(x) \tag{3.9}$$

and assume that $\widehat{f}(\alpha)$ is even in $\alpha$. Next, we remark that the Jacobi functions obey the following connection identity:

$$\vartheta_{\alpha}^{(p,q)}(x) = \frac{1}{2} \left[ Q_{p,q}(\alpha) G_{\alpha}^{(p,q)}(x) + Q_{p,q}(-\alpha) G_{-\alpha}^{(p,q)}(x) \right] \tag{3.10}$$

with

$$G_{\alpha}^{(p,q)}(x) = x^{\frac{1}{2}(1+p+q)+\alpha} \, {}_2F_1 \left( \begin{matrix} \frac{1}{2}(1+p+q) + \alpha, \, \frac{1}{2}(1+p-q) + \alpha \\ 1 + 2\alpha \end{matrix} ; x \right). \tag{3.11}$$

For definite values of $p$ and $q$, the functions $G_{\alpha}^{(p,q)}(x)$ will be identified with conformal blocks of scaling dimension $\Delta \sim \alpha$. At this point, we can replace $\vartheta_{\alpha}^{(p,q)} \to Q_{p,q}(\alpha) G_{\alpha}^{(p,q)}$ in the integrand (3.9), whence

$$f(x) = 2 \int_{\mathcal{C}} \frac{[d\alpha]}{Q_{p,q}(-\alpha)} \, \widehat{f}(\alpha) G_{\alpha}^{(p,q)}(x). \tag{3.12}$$

Closing the contour $\mathcal{C}$, we can recast $f(x)$ as a sum over all poles $\alpha_n$ circled by $\mathcal{C}$:

$$f(x) = \sum_n c_n \, G_{\alpha_n}^{(p,q)}(x), \quad c_n = -\frac{2}{Q_{p,q}(-\alpha_n)} \operatorname{Res} \widehat{f}(\alpha) \big|_{\alpha_n}. \tag{3.13}$$

In CFT examples, the poles $\alpha_n$ will play the role of scaling dimensions, and the residues $c_n$ will be related to OPE coefficients.

We conclude this section with two useful facts about the parameters $p, q$ of the Jacobi operator $\mathcal{D}_{p,q}$. If we change the sign of either $p$ or $q$, this operator transforms in a simple way, namely

$$\mathcal{D}_{-p,q} = \left( \frac{1-x}{x} \right)^p \cdot [\mathcal{D}_{p,q} + p(1+q)] \cdot \left( \frac{x}{1-x} \right)^p \tag{3.14a}$$

$$\mathcal{D}_{p,-q} = x^{-q} \cdot [\mathcal{D}_{p,q} + (1+p)q] \cdot x^q. \tag{3.14b}$$

The first of these identities can be used to find a second continuous family of eigenfunctions of $\mathcal{D}_{p,q}$, i.e.

$$\left( \frac{x}{1-x} \right)^p \vartheta_{\alpha}^{(-p,q)}(x), \quad \alpha \in i\mathbb{R}. \tag{3.15}$$

These new solutions are linearly independent from the Jacobi functions $\vartheta_{\alpha}^{(p,q)}(x)$, provided that $p \neq 0$. The second identity (3.14b) does not give rise to new eigenfunctions, due to the functional identity

$$\vartheta_{\alpha}^{(p,q)}(x) = x^q \, \vartheta_{\alpha}^{(p,-q)}(x). \tag{3.16}$$

# 4 Boundary CFTs

Let us now turn to a first example, that of a $d$-dimensional boundary BCFT. We have in mind a theory defined on half of Euclidean space, with a boundary at the hyperplane $x^d = 0$. To fix the notation, we can label coordinates parallel to the boundary as $\mathbf{x}$, hence points in the "bulk" are parametrized as $x^\mu = (\mathbf{x}, x^d)$. For concreteness we will consider two-point functions of scalar primary operators, although *a priori* it is possible to consider more general two-point functions.

## 4.1 Boundary bootstrap equation

As announced, we will consider a two-point function $\langle \mathcal{O}_1(x)\mathcal{O}_2(y)\rangle$ in a boundary CFT, where the $\mathcal{O}_i$ are scalar primary operators of dimension $\Delta_i$. Such correlators have been studied in great detail [66–68]: in this section, we will briefly review some classic results without adding any new material. The same material is reviewed more extensively in the bootstrap papers [69, 70].

Let us briefly consider conformal kinematics in the presence of boundary. The presence of a boundary breaks the conformal group down to a subgroup $SO(d, 1)$, e.g. translations in the $x^d$-direction are no longer symmetries. This means that $n$-point functions are less constrained than in the usual flat-space setting [89]. Taking a one-point function $\langle \mathcal{O}_i(x)\rangle$ for concreteness, the $SO(d, 1)$ subgroup restricts it to have the following functional form:

$$\langle \mathcal{O}_i(x)\rangle = \frac{\mu_i}{(2x^d)^{\Delta_i}} \tag{4.1}$$

where $\mu_i$ is an arbitrary real number, not fixed by symmetry arguments. In general, a CFT can have several different boundary conditions. Changing the boundary condition amounts to changing the coefficients $\{\mu_i\}$.

Next, consider the two-point function $\langle \mathcal{O}_1(x)\mathcal{O}_2(y)\rangle$ of interest. Given two points $x, y$, we remark that the following ratio

$$\rho = \frac{(x - y)^2}{4x^d y^d + (x - y)^2} \in [0, 1] \tag{4.2}$$

is conformally invariant.[4] This means that conformal symmetry only determines the form of the correlator $\langle \mathcal{O}_1\mathcal{O}_2\rangle$ up to an arbitrary function $\mathcal{F}(\rho)$. To be precise, we can write the two-point function of interest as

$$\langle \mathcal{O}_1(x)\mathcal{O}_2(y)\rangle = \frac{\mathcal{F}(\rho)}{(2x^d)^{\Delta_1}(2y^d)^{\Delta_2}} \,. \tag{4.3}$$

The kinematics of $\rho$ are as follows: $\rho \to 0$ corresponds to inserting the two operators $\mathcal{O}_{1,2}$ close to one another, whereas $\rho \to 1$ corresponds to bringing at least one of the operators close to the boundary.

In a boundary CFT, there exist two inequivalent ways to compute the correlation function $\mathcal{F}(\rho)$. The first one makes use of the normal, or bulk, OPE:

$$\mathcal{O}_1(x)\mathcal{O}_2(y) \sim \sum_k \lambda_{12}{}^k \mathcal{O}_k(y) + \text{descendants} \tag{4.4}$$

---

[4]This $\rho$-coordinate is unrelated to the one introduced in [90] or [43]. The commonly used [68] BCFT cross-ratio $\xi \in [0, \infty)$ is related to $\rho$ through $\rho = \xi/(\xi + 1)$.

where the sum runs over all bulk primaries $\mathcal{O}_k$. Here the $\lambda_{12}{}^k$ are the standard OPE coefficients, proportional to the flat-space three-point function $\langle \mathcal{O}_1 \mathcal{O}_2 \mathcal{O}_k \rangle$. By resumming the bulk OPE (4.4), one can derive [68] the following expansion for $\mathcal{F}$:

$$\mathcal{F}(\rho) = \left( \frac{1-\rho}{\rho} \right)^{\Delta_+} \sum_k \lambda_{12}{}^k \mu_k \, G_{\Delta_k}^{\text{bulk}}(\rho) , \quad \Delta_+ := \frac{\Delta_1 \pm \Delta_2}{2} . \tag{4.5}$$

The above sum runs over all scalar primaries $\mathcal{O}_k$ with dimension $\Delta_k$; the coefficients $\mu_k$ are the VEVs from Eq. (4.1), and the functions $G_\Delta^{\text{bulk}}$ are bulk conformal blocks:

$$G_\Delta^{\text{bulk}}(\rho) = \rho^{\Delta/2}(1-\rho)^{\Delta_-} \, {}_2F_1 \left( \frac{\Delta}{2} + \Delta_-, \frac{\Delta}{2} + 1 - h + \Delta_-; \Delta + 1 - h; \rho \right) . \tag{4.6}$$

Eq. (4.5) is known as a bulk conformal block (CB) decomposition.

However, the correlator $\mathcal{F}(\rho)$ can also be computed in a different way. This time, we rely on the fact that the boundary theory is a CFT in its own right, and its spectrum is described by a set of boundary primaries $O_j(\mathbf{x})$. Applying the state-operator correspondance to the boundary CFT, it follows that bulk operators satisfy a boundary OPE, schematically:

$$\mathcal{O}_1(x) \sim \sum_j b_{1j} O_j(\mathbf{x}) + \text{descendants} . \tag{4.7}$$

The sum on the RHS runs over all scalar primaries of the boundary theory. We note that the coefficients $\{b_{ij}\}$ depend on the boundary condition in question, as was the case for the VEVs $\{\mu_i\}$. Evidently, the bulk primary $\mathcal{O}_2(y)$ admits a similar boundary decomposition. By carefully resumming the expansion (4.7), we can therefore derive a boundary CB decomposition for $\mathcal{F}(\rho)$:

$$\mathcal{F}(\rho) = \sum_j b_{1j} b_{2j} \, G_{\Delta_j}^{\text{bdy}}(\rho) \tag{4.8}$$

where the boundary conformal blocks are given by

$$G_\Delta^{\text{bdy}}(\rho) = (1-\rho)^\Delta \, {}_2F_1 \left( \begin{matrix} \Delta, \Delta + 1 - h \\ 2\Delta + 2 - 2h \end{matrix} ; 1 - \rho \right) . \tag{4.9}$$

Requiring that the bulk (4.5) and boundary (4.8) decompositions agree, we arrive at the following consistency condition:

$$\sum_k \lambda_{12}{}^k \mu_k \, G_{\Delta_k}^{\text{bulk}}(\rho) = \left( \frac{\rho}{1-\rho} \right)^{\Delta_+} \sum_j b_{1j} b_{2j} \, G_{\Delta_j}^{\text{bdy}}(\rho) \tag{4.10}$$

which converges for all $0 < \rho < 1$. Eq. (4.10) is known as a boundary bootstrap equation. It can be used to simultaneously constrain the coefficients $\{\mu_i\}$ and $\{b_{ij}\}$, as has been done in [69, 70]. From now on, we will mostly be interested in formal aspects of (4.10), rather than in finding numerical constraints on these parameters.

## 4.2 Sturm-Liouville theory in the bulk...

In line with the Jacobi transform discussion from Sec. 3, we aim to develop an integral decomposition for the correlator $\mathcal{F}(\rho)$. As a starting point, we notice that the bulk blocks are solutions to a Casimir differential equation:

$$D_{\text{bulk}} \cdot G_\Delta^{\text{bulk}}(\rho) = C_d(\Delta) G_\Delta^{\text{bulk}}(\rho) \tag{4.11}$$

where

$$\frac{1}{4} D_{\text{bulk}} \cdot f(\rho) = w_{\text{bulk}}(\rho)^{-1} \frac{d}{d\rho} \left[ \rho^{1-h}(1-\rho) f'(\rho) \right] - \Delta_-^2 \frac{\rho}{1-\rho} f(\rho), \quad w_{\text{bulk}}(\rho) = \frac{1}{\rho^{h+1}}. \tag{4.12}$$

The suggestive form of (4.12) implies that $D_{\text{bulk}}$ is self-adjoint with respect to the inner product

$$\langle f, g \rangle_{\text{bulk}} = \int_0^1 d\rho \, w_{\text{bulk}}(\rho) \, \overline{f(\rho)} g(\rho) \tag{4.13}$$

which depends only on the spacetime dimension $d$. We will denote the space of square-normalizable functions with respect to (4.13) as $\mathfrak{H}_{\text{bulk}}$.

At this stage we would like to make contact with the Sturm-Liouville theory of the Jacobi operator $\mathcal{D}_{p,q}$ from Sec. 3. If $\Delta_1 = \Delta_2$ (i.e. $\Delta_- = 0$) this is easy: we recognize immediately that

$$\frac{1}{4} D_{\text{bulk}} = \mathcal{D}_{0,h-1} \qquad (x = \rho, \ \Delta_1 = \Delta_2). \tag{4.14}$$

In the general case ($\Delta_1 \neq \Delta_2$), we recover the Jacobi operator $\mathcal{D}_{p,q}$ with parameters $(p,q) = (2\Delta_-, h-1)$ after gauging away a simple prefactor. The precise identification is given by

$$\frac{1}{4} D_{\text{bulk}} = \left( \frac{1-\rho}{\rho} \right)^{\Delta_-} \cdot \left[ \mathcal{D}_{2\Delta_-, h-1} + \Delta_-(\Delta_- + h) \right] \cdot \left( \frac{\rho}{1-\rho} \right)^{\Delta_-} \qquad (x = \rho). \tag{4.15}$$

This means that we can use the properties of the Jacobi functions to find a complete basis of eigenfunctions of $D_{\text{bulk}}$, to wit:

$$\Psi_\alpha^{\text{bulk}}(\rho) = \left( \frac{1-\rho}{\rho} \right)^{\Delta_-} \vartheta_\alpha^{(2\Delta_-, h-1)}(\rho), \quad \alpha \in i\mathbb{R}. \tag{4.16}$$

We will refer to the basis functions $\Psi_\alpha^{\text{bulk}}(\rho)$ as *bulk partial waves*. The appropriate boundary alpha space transform is thus

$$\boxed{f(\rho) = \int \frac{[d\alpha]}{N_{\text{bulk}}(\alpha)} \, \widehat{f}(\alpha) \Psi_\alpha^{\text{bulk}}(\rho) \ \leftrightarrow \ \widehat{f}(\alpha) = \int_0^1 d\rho \, w_{\text{bulk}}(\rho) \, f(\rho) \Psi_\alpha^{\text{bulk}}(\rho)} \tag{4.17}$$

where

$$N_{\text{bulk}}(\alpha) = \frac{|Q_{\text{bulk}}(\alpha)|^2}{2}, \quad Q_{\text{bulk}}(\alpha) = Q_{2\Delta_-, h-1}(\alpha). \tag{4.18}$$

In terms of Hilbert spaces, the alpha space transform $f(\rho) \mapsto \widehat{f}(\alpha)$ from (4.17) induces a map $\mathfrak{H}_{\text{bulk}} \to \mathfrak{A}_{2\Delta_-, h-1}$.

In particular, we can apply (4.17) to the two-point function $\mathcal{F}(\rho)$:

$$\mathcal{F}(\rho) = \left(\frac{1-\rho}{\rho}\right)^{\Delta_+} \int_{\mathcal{C}} \frac{[d\alpha]}{N_{\text{bulk}}(\alpha)} \mathcal{F}_{\text{bulk}}(\alpha) \Psi_\alpha^{\text{bulk}}(\rho) \tag{4.19}$$

which defines a bulk spectral density $\mathcal{F}_{\text{bulk}}(\alpha)$. To check that (4.19) reproduces the bulk CB decomposition (4.5), we use that $\Psi_\alpha^{\text{bulk}}$ is the sum of a bulk CB and its shadow:

$$\Psi_\alpha^{\text{bulk}}(\rho) = \frac{1}{2}\left[Q_{\text{bulk}}(\alpha)G_{h+2\alpha}^{\text{bulk}}(\rho) + (\alpha \to -\alpha)\right] \tag{4.20}$$

as follows from Eq. (3.10). Using the contour trick described in Sections 2 and 3, we indeed establish that

$$\mathcal{F}(\rho) = \left(\frac{1-\rho}{\rho}\right)^{\Delta_+} \sum_n c_n G_{h+2\alpha_n}^{\text{bulk}}(\rho), \quad c_n = -\frac{2}{Q_{\text{bulk}}(-\alpha_n)} \text{Res } \mathcal{F}_{\text{bulk}}(\alpha)\big|_{\alpha=\alpha_n}, \tag{4.21}$$

where the sum runs over all poles $\alpha_n$ of $\mathcal{F}_{\text{bulk}}(\alpha)$, circled by $\mathcal{C}$ in the positive direction.

We will finish with two technical remarks. First, using (4.20) one sees that the representation (4.19) is really an integral over conformal blocks with scaling dimension $h + 2\alpha = d/2 + 2\alpha$. This points to a group-theoretic interpretation, as the family $\Delta = d/2 + i\mathbb{R}$ forms the unitary principal series of $SO(d+1,1)$, restricted to the scalar sector. Our integral representation for $\mathcal{F}(\rho)$ is therefore very similar to the conformal partial wave representations for flat-space correlators derived in e.g. [91, 47].

Second, we notice that the blocks $G_\Delta^{\text{bulk}}$ are invariant under the exchange $\Delta_1 \leftrightarrow \Delta_2$, but the bulk partial waves $\Psi_\alpha^{\text{bulk}}$ do not have this symmetry. In fact, we made an arbitrary choice by using the functions $\Psi_\alpha^{\text{bulk}}$ as a basis. One could equally well use a different basis, given by the functions

$$\Psi_\alpha'(\rho) = \Psi_\alpha^{\text{bulk}}(\rho)\big|_{\Delta_1 \leftrightarrow \Delta_2} \tag{4.22}$$

cf. Eq. (3.15). This would lead to a new alpha space transform, which differs only in having $\Delta_1$ and $\Delta_2$ exchanged in all formulas.

## 4.3   . . . and on the boundary

Let us proceed by developing a second Sturm-Liouville decomposition for $\mathcal{F}(\rho)$, one that is adapted to the boundary blocks (4.9). The logic used will be completely identical. We start by remarking that the boundary blocks are eigenfunctions — with eigenvalue $C_{d-1}(\Delta)$ — of the differential operator

$$D_{\text{bdy}} \cdot f(\rho) = w_{\text{bdy}}(\rho)^{-1} \frac{d}{d\rho}\left[\rho(1-\rho)^2 w_{\text{bdy}}(\rho) f'(\rho)\right], \quad w_{\text{bdy}}(\rho) = \frac{\rho^{h-1}}{(1-\rho)^{2h}}. \tag{4.23}$$

It follows that $D_{\text{bdy}}$ is self-adjoint with respect to the inner product

$$\langle f, g \rangle_{\text{bdy}} = \int_0^1 d\rho\, w_{\text{bdy}}(\rho) \overline{f(\rho)} g(\rho) \tag{4.24}$$

and we will denote the corresponding Hilbert space as $\mathfrak{H}_{\text{bdy}}$.

In order to find a basis of $\mathfrak{H}_{\text{bdy}}$, we once again identify $D_{\text{bdy}}$ with the Jacobi operator $\mathcal{D}_{p,q}$, this time with parameters $p = q = h - 1$:

$$D_{\text{bdy}} = \mathcal{D}_{h-1,\,h-1} \qquad (x = 1 - \rho). \tag{4.25}$$

Consequently, a complete basis of eigenfunctions is given by the *boundary partial waves*

$$\Psi_\nu^{\text{bdy}}(\rho) = \vartheta_\nu^{(h-1,\,h-1)}(1 - \rho), \quad \nu \in i\mathbb{R}. \tag{4.26}$$

Therefore, any function $f(\rho) \in \mathfrak{H}_{\text{bdy}}$ can be decomposed as follows:

$$\boxed{f(\rho) = \int \frac{[d\nu]}{N_{\text{bdy}}(\nu)} \, \widehat{f}(\nu) \Psi_\nu^{\text{bdy}}(\rho) \; \leftrightarrow \; \widehat{f}(\nu) = \int_0^1 d\rho \, w_{\text{bdy}}(\rho) \, f(\rho) \Psi_\nu^{\text{bdy}}(\rho)} \tag{4.27}$$

where

$$N_{\text{bdy}}(\nu) = \frac{|Q_{\text{bdy}}(\nu)|^2}{2}, \quad Q_{\text{bdy}}(\nu) = Q_{h-1,\,h-1}(\nu). \tag{4.28}$$

In the Hilbert space language, the map $f(\rho) \mapsto \widehat{f}(\nu)$ defines an isometry $\mathfrak{H}_{\text{bdy}} \to \mathfrak{A}_{h-1,h-1}$.

As before, we can recover the boundary CB decomposition (4.8) from the boundary transform (4.27). We start by writing

$$\mathcal{F}(\rho) = \int_{\mathcal{C}'} \frac{[d\nu]}{N_{\text{bdy}}(\nu)} \, \mathcal{F}_{\text{bdy}}(\nu) \Psi_\nu^{\text{bdy}}(\rho) \tag{4.29}$$

which defines a boundary alpha space density $\mathcal{F}_{\text{bdy}}(\nu)$. We stress that the contour $\mathcal{C}'$ appearing here is not related to the contour $\mathcal{C}$ from (4.19). Next, we make use of the connection formula

$$\Psi_\nu^{\text{bdy}}(\rho) = \frac{1}{2} \left[ Q_{\text{bdy}}(\nu) G_{h-\frac{1}{2}+\nu}^{\text{bdy}}(\rho) + (\nu \to -\nu) \right]. \tag{4.30}$$

Closing the contour, we therefore find that $\mathcal{F}(\rho)$ can be expressed as follows:

$$\mathcal{F}(\rho) = \sum_j c_j G_{h-\frac{1}{2}+\nu_j}^{\text{bdy}}(\rho), \quad c_j = -\frac{2}{Q_{\text{bdy}}(-\nu_j)} \left. \text{Res} \, \mathcal{F}_{\text{bdy}}(\nu) \right|_{\nu=\nu_j}, \tag{4.31}$$

where the sum runs over all poles $\nu_j$ circled by $\mathcal{C}'$.

We finish by pointing to the group theory interpretation of the spectral representation (4.29). This time, the integral runs over all scalar representations of dimension $h - \frac{1}{2} + i\mathbb{R} = \frac{1}{2}(d-1) + i\mathbb{R}$. This is precisely what one could have expected, as this is the scalar part of the principal series of $SO(d, 1)$, befitting a CFT in $d - 1$ spacetime dimensions.

### 4.3.1 Intermezzo: conformal Regge theory

The boundary partial waves described above have appeared in the CFT literature before, albeit in a different guise, in the context of conformal Regge theory [92, 93, 39]. In this section we will briefly spell out the connection. In conformal Regge theory one is interested in a four-point function $\mathcal{A}(u, v)$ in a large-$N$ CFT, which maps to a $2 \to 2$ scattering amplitude in $\text{AdS}_{d+1}$ in a way that can be made precise. Regge kinematics correspond to replacing $(u, v)$ by two new cross

ratios $(s, r)$ and taking the limit $s \to 0$ at fixed $r$.[5] A key result — formula (56) of [39] — is that in this regime the correlator $\mathcal{A}(s, r)$ can be decomposed as

$$\mathcal{A}(s, r) \approx \int_{\mathbb{R}} d\nu\, R(s, \nu)\, \Omega_{i\nu}^{(d-1)}(r) \tag{4.32}$$

where $R(s, \nu)$ captures information about the leading Regge trajectory and the $\Omega_{i\nu}^{(d)}(r)$ are "radial" eigenfunctions of the Laplacian on $d$-dimensional hyperbolic space. It is an ancient result that the functions $\Omega_{i\nu}^{(d)}$ can be expressed in terms of Jacobi functions [94]. The precise relation between $\Omega_{i\nu}^{(d)}$ and the boundary partial waves is

$$\Omega_{i\nu}^{(d)}(r) = \frac{\Gamma(d/2)}{2^{d+1}\pi^{d/2+1}} \frac{1}{N_{\mathrm{bdy}}(i\nu)} \Psi_{i\nu}^{\mathrm{bdy}}\big(\tanh^2(r/2)\big) \tag{4.33}$$

as can be established using an explicit formula for $\Omega_{i\nu}^{(d)}$ given in Ref. [39]. Eq. (4.33) shows that the integral transform $R(s, \nu) \mapsto \mathcal{A}(s, r)$ from (4.32) is identical to the inverse boundary alpha space transform (4.27), after shifting the spacetime dimension $d$ by one unit, performing a coordinate change $\nu \to i\nu$ and absorbing an unimportant prefactor into the measure $N_{\mathrm{bdy}}(i\nu)$. Although we will not push the analogy between Regge amplitudes and BCFT two-point correlators any further, it would be interesting to understand the origin of (4.33) in more detail.

## 4.4 Crossing kernel

Let us summarize our results so far. We have derived two different integral representations for a two-point function $\langle \mathcal{O}_1(x)\mathcal{O}_2(y)\rangle \sim \mathcal{F}(\rho)$ in a boundary CFT. The spectral densities appearing in these representations encode bulk resp. boundary CFT data.

The logical next step is to enforce that these different representations agree:

$$\int \frac{[d\alpha]}{N_{\mathrm{bulk}}(\alpha)}\, \mathcal{F}_{\mathrm{bulk}}(\alpha)\Psi_{\alpha}^{\mathrm{bulk}}(\rho) = \left(\frac{\rho}{1-\rho}\right)^{\Delta_+} \int \frac{[d\nu]}{N_{\mathrm{bdy}}(\nu)}\, \mathcal{F}_{\mathrm{bdy}}(\nu)\Psi_{\nu}^{\mathrm{bdy}}(\rho)\,. \tag{4.34}$$

This is the alpha space version of the position space bootstrap equation (4.10). The consistency condition (4.34) can be graphically represented, see Fig. 1.

**Figure 1:** Schematic representation of the boundary bootstrap equation in alpha space.

It is clear that (4.34) imposes constraints on the spectral densities $\mathcal{F}_{\mathrm{bulk}}(\alpha)$ and $\mathcal{F}_{\mathrm{bdy}}(\nu)$. To make these constraints transparent we can use the completeness of the bulk and boundary partial waves and expand one in terms of the other. For instance, we can represent

$$\left(\frac{\rho}{1-\rho}\right)^{\Delta_+} \Psi_{\nu}^{\mathrm{bdy}}(\rho) = \int \frac{[d\alpha]}{N_{\mathrm{bulk}}(\alpha)}\, E_{\mathrm{bdy}\to\mathrm{bulk}}(\alpha, \nu)\Psi_{\alpha}^{\mathrm{bulk}}(\rho)\,. \tag{4.35}$$

---

[5]Ref. [39] uses the notation $(\sigma, \rho)$ instead of $(s, r)$.

This equation defines a set of coefficients $E_{\text{bdy}\to\text{bulk}}(\alpha, \nu)$, depending on the external dimensions $\Delta_1$ and $\Delta_2$ and the spacetime dimension $d$. Likewise, we can write down a decomposition of bulk partial waves in terms of boundary waves:

$$\left(\frac{1-\rho}{\rho}\right)^{\Delta_+} \Psi_\alpha^{\text{bulk}}(\rho) = \int \frac{[d\nu]}{N_{\text{bdy}}(\nu)} E_{\text{bulk}\to\text{bdy}}(\nu, \alpha) \Psi_\nu^{\text{bdy}}(\rho) \tag{4.36}$$

defining a second set of coefficients $E_{\text{bulk}\to\text{bdy}}(\nu, \alpha)$. Once more, these equations can be schematically represented, see Fig. 2.

**Figure 2:** Boundary (resp. bulk) partial waves can be expressed in terms of bulk (boundary) ones by means of the kernels $E_{\text{bdy}\to\text{bulk}}$ and $E_{\text{bulk}\to\text{bdy}}$.

Using standard CFT jargon we will refer to $E_{\text{bdy}\to\text{bulk}}$ and $E_{\text{bulk}\to\text{bdy}}$ as *crossing kernels*, although strictly speaking they have nothing to do with Bose or crossing symmetry — it is easier to think of them as change-of-basis matrices. The two kernels are useful in recasting the consistency condition (4.34). For instance, we can plug (4.35) into (4.34), which yields an integral equation:

$$\mathcal{F}_{\text{bulk}}(\alpha) = (\mathsf{E}_{\text{bdy}\to\text{bulk}} \cdot \mathcal{F}_{\text{bdy}})(\alpha) \tag{4.37}$$

where we have introduced an integral operator $\mathsf{E}_{\text{bdy}\to\text{bulk}}$ as follows:

$$(\mathsf{E}_{\text{bdy}\to\text{bulk}} \cdot f)(\alpha) := \int \frac{[d\nu]}{N_{\text{bdy}}(\nu)} E_{\text{bdy}\to\text{bulk}}(\alpha, \nu) f(\nu) \,. \tag{4.38}$$

We can also invert the logic, inserting Eq. (4.36) into the alpha space bootstrap equation. This leads to a second integral equation, namely

$$\mathcal{F}_{\text{bdy}}(\nu) = (\mathsf{E}_{\text{bulk}\to\text{bdy}} \cdot \mathcal{F}_{\text{bulk}})(\nu) \,, \quad (\mathsf{E}_{\text{bulk}\to\text{bdy}} \cdot f)(\nu) := \int \frac{[d\alpha]}{N_{\text{bulk}}(\alpha)} E_{\text{bulk}\to\text{bdy}}(\nu, \alpha) f(\alpha) \,. \tag{4.39}$$

Summarizing, equality of the bulk and boundary operator expansions leads to a $2 \times 2$ system of integral equations for the relevant spectral densities:

$$\boxed{\mathcal{F}_{\text{bulk}}(\alpha) = (\mathsf{E}_{\text{bdy}\to\text{bulk}} \cdot \mathcal{F}_{\text{bdy}})(\alpha) \quad \text{and} \quad \mathcal{F}_{\text{bdy}}(\nu) = (\mathsf{E}_{\text{bulk}\to\text{bdy}} \cdot \mathcal{F}_{\text{bulk}})(\nu) \,.} \tag{4.40}$$

Eq. (4.40) is one of the key results of this paper. The actual kernels $E_{\text{bdy}\to\text{bulk}}$ and $E_{\text{bulk}\to\text{bdy}}$ can be computed using the alpha space technology of Secs. 4.2 and 4.3, yielding the following integral expressions:

$$E_{\text{bdy}\to\text{bulk}}(\alpha, \nu) = \int_0^1 d\rho \, w_{\text{bulk}}(\rho) \left(\frac{\rho}{1-\rho}\right)^{\Delta_+} \Psi_\alpha^{\text{bulk}}(\rho) \Psi_\nu^{\text{bdy}}(\rho) \,, \tag{4.41a}$$

$$E_{\text{bulk}\to\text{bdy}}(\nu, \alpha) = \int_0^1 d\rho \, w_{\text{bdy}}(\rho) \left(\frac{1-\rho}{\rho}\right)^{\Delta_+} \Psi_\alpha^{\text{bulk}}(\rho) \Psi_\nu^{\text{bdy}}(\rho) \,. \tag{4.41b}$$

Starting from Eq. (4.41), it is possible to derive explicit formulas for $E_{\text{bdy}\to\text{bulk}}$ and $E_{\text{bulk}\to\text{bdy}}$. The computation in question would be similar to the one performed in [1] for the $SL(2,\mathbb{R})$ crossing kernel. We will postpone this problem to Sec. 6.

Let us conclude by exhibiting a duality relation that $E_{\text{bdy}\to\text{bulk}}$ and $E_{\text{bulk}\to\text{bdy}}$ obey, similar to Eq. (2.20) in the toy example. Making the dependence of these kernels on the external dimensions $\Delta_1$ and $\Delta_2$ explicit, we have

$$E_{\text{bdy}\to\text{bulk}}(\alpha,\nu|\Delta_1,\Delta_2) = E_{\text{bulk}\to\text{bdy}}(\nu,\alpha|d-\Delta_2,d-\Delta_1). \tag{4.42}$$

This can be shown by inspecting the integrands in (4.41).

# 5 Crosscap CFTs

In the next part of this paper we consider CFTs defined on $\mathbb{RP}^d$, the $d$-dimensional counterpart of the crosscap. As in the boundary case, we will specialize to scalar two-point functions, and our discussion will mimic the BCFT case to a certain extent. We will start by reviewing CFT kinematics on the crosscap and describing the relevant conformal block decomposition, before developing the appropriate alpha space transform and obtaining a crosscap consistency condition in alpha space. For a more comprehensive discussion of CFTs on the crosscap, we refer to [81, 78, 82].

## 5.1 Bootstrap condition

In this section we will review some facts about scalar two-point functions on real projective space, without contributing any new results. The compact manifold $\mathbb{RP}^d$ is defined as a quotient $S^d/\sim$, obtained by identifying antipodal points on the $d$-sphere. It is convenient to map $S^d$ to Euclidean space via the stereographic projection. If we parametrize points on $\mathbb{R}^d$ as $x^\mu$, the crosscap equivalence relation reads

$$x^\mu \sim -\frac{x^\mu}{x^2} \equiv \tilde{x}^\mu. \tag{5.1}$$

As a fundamental domain of $\mathbb{RP}^d$ we can e.g. choose the unit ball $|x| \leq 1$ with antipodal points on the boundary $|x| = 1$ identified. The equivalence relation (5.1) breaks the conformal group down to a subgroup $SO(d,1)$, similar to the situation in the boundary case. Scalar one-point functions are again allowed to be non-zero; if $\mathcal{O}_i$ is a scalar primary of dimension $\Delta_i$, then its one-point function reads

$$\langle \mathcal{O}_i(x) \rangle = \frac{a_i}{(1+x^2)^{\Delta_i}}. \tag{5.2}$$

The coefficients $a_i$ parametrize a set of crosscap CFT data — in particular, these numbers cannot be probed using flat-space correlation functions.

We will be interested in the two-point function $\langle \mathcal{O}_1(x)\mathcal{O}_2(y) \rangle$ of two scalar primaries. This correlator is not completely fixed by conformal symmetry, as

$$\eta = \frac{(x-y)^2}{(1+x^2)(1+y^2)} \in [0,1] \tag{5.3}$$

is an $SO(d,1)$ invariant cross ratio. The most general form the correlator can take is therefore

$$\langle \mathcal{O}_1(x)\mathcal{O}_2(y) \rangle = \frac{\eta^{-\Delta_+}\mathcal{G}(\eta)}{(1+x^2)^{\Delta_1}(1+y^2)^{\Delta_2}} \tag{5.4}$$

where $\mathcal{G}(\eta)$ is not constrained by conformal kinematics. Using the OPE (4.4), it can be shown that this function admits a CB decomposition similar to Eq. (4.5). In the present case, we learn that $\mathcal{G}(\eta)$ can be written as a sum over all scalar primaries $\mathcal{O}_k$ in the theory, with coefficients that depend on the coefficients $a_k$ from (5.2):

$$\mathcal{G}(\eta) = \sum_k \lambda_{12}{}^k a_k \, G_{\Delta_k}^{\mathrm{proj}}(\eta) \, . \tag{5.5}$$

The crosscap conformal blocks $G_\Delta^{\mathrm{proj}}(\eta)$ appearing here are given by

$$G_\Delta^{\mathrm{proj}}(\eta) = \eta^{\Delta/2} \, {}_2F_1\left(\frac{\Delta}{2} + \Delta_-, \frac{\Delta}{2} - \Delta_-; \Delta + 1 - h; \eta\right), \tag{5.6}$$

and we note that (5.15) converges for all $0 \le \eta \le 1$.

The crux of the crosscap bootstrap is that the coefficients $\{a_i\}$ are not arbitrary; rather, they are constrained by an equation similar to (4.10). To prove this, one notices that $\mathcal{G}(\eta)$ must obey a non-trivial functional identity. The reason is that any two points $x$ and $\tilde{x}$ must be identified, yet the cross ratio $\eta$ is *not* invariant under the involution $x \mapsto \tilde{x}$ (or $y \mapsto \tilde{y}$): it transforms as $\eta \mapsto 1 - \eta$. Imposing that the correlation function $\langle \mathcal{O}_1 \mathcal{O}_2 \rangle$ is consistent therefore leads to the following bootstrap condition:

$$\mathcal{G}(\eta) = \left(\frac{\eta}{1-\eta}\right)^{\Delta_+} \mathcal{G}(1-\eta) \tag{5.7}$$

or equivalently

$$\sum_k \lambda_{12}{}^k a_k \left[ G_{\Delta_k}^{\mathrm{proj}}(\eta) - \left(\frac{\eta}{1-\eta}\right)^{\Delta_+} G_{\Delta_k}^{\mathrm{proj}}(1-\eta) \right] = 0 \, . \tag{5.8}$$

Since (5.8) is not satisfied term-by-term, it imposes a simultaneous constraint on the spectrum $\{\Delta_k\}$ of the CFT and the coefficients $\lambda_{12}{}^k a_k$.

It turns out that the bootstrap equation Eq. (5.7) is slightly too constraining. The reason is that operators $\mathcal{O}_i(x)$ can pick up a sign $\epsilon_i = \pm 1$ under the involution $x \mapsto \tilde{x}$. However, by moving both $\mathcal{O}_1(x)$ and $\mathcal{O}_2(y)$ to their mirror points, we see that $\langle \mathcal{O}_1 \mathcal{O}_2 \rangle$ vanishes unless $\epsilon_1 = \epsilon_2$. In general, we conclude that the function $\mathcal{G}(\eta)$ is either symmetric or antisymmetric under crossing, depending on the sign of $\epsilon_1 = \epsilon_2$.

## 5.2 Sturm-Liouville theory for the crosscap

We now take the CB decomposition (5.15) as a starting point to derive an alpha space transform for $d$-dimensional crosscap correlators. By now, the logic will be familiar. As a starting point we notice that the blocks $G_\Delta^{\mathrm{proj}}(\eta)$ are eigenfunctions of a second-order Casimir operator:

$$\frac{1}{4} D_{\mathrm{proj}} \cdot f(\eta) = w_{\mathrm{proj}}(\eta)^{-1} \frac{d}{d\eta}\left[ (1-\eta)\eta^2 w_{\mathrm{proj}}(\eta) f'(\eta) \right] + \Delta_-^2 \, \eta f(\eta) \, , \quad w_{\mathrm{proj}}(\eta) = \frac{(1-\eta)^{h-1}}{\eta^{h+1}} \, . \tag{5.9}$$

Thus $D_{\mathrm{proj}}$ is self-adjoint with respect to the inner product

$$\langle f, g \rangle_{\mathrm{proj}} = \int_0^1 d\eta \, w_{\mathrm{proj}}(\eta) \, \overline{f(\eta)} g(\eta) \tag{5.10}$$

and we denote the relevant Hilbert space by $\mathfrak{H}_{\text{proj}}$. Next, we write $D_{\text{proj}}$ in terms of the Jacobi operator $\mathcal{D}_{p,q}$ as follows:

$$\frac{1}{4}D_{\text{proj}} = \frac{1}{\eta^{\Delta_-}} \cdot \left[\mathcal{D}_{h-1,\,2\Delta_-} + \Delta_-(\Delta_- + h)\right] \cdot \eta^{\Delta_-} \qquad (x = \eta)\,. \tag{5.11}$$

Consequently, a complete basis of functions is given by the *crosscap partial waves*

$$\Phi_\alpha(\eta) = \eta^{-\Delta_-}\,\vartheta_\alpha^{(h-1,\,2\Delta_-)}(\eta)\,, \qquad \alpha \in i\mathbb{R}\,. \tag{5.12}$$

The appropriate crosscap alpha space transform is therefore

$$\boxed{f(\eta) = \int \frac{[d\alpha]}{N_{\text{proj}}(\alpha)}\,\widehat{f}(\alpha)\Phi_\alpha(\eta) \;\leftrightarrow\; \widehat{f}(\alpha) = \int_0^1 d\eta\, w_{\text{proj}}(\eta)\,f(\eta)\Phi_\alpha(\eta)} \tag{5.13}$$

where

$$N_{\text{proj}}(\alpha) = \frac{|Q_{\text{proj}}(\alpha)|^2}{2}\,, \qquad Q_{\text{proj}}(\alpha) = Q_{h-1,\,2\Delta_-}\,, \tag{5.14}$$

and $f(\eta) \mapsto \widehat{f}(\alpha)$ induces a unitary map $\mathfrak{H}_{\text{proj}} \to \mathfrak{A}_{h-1,\,2\Delta_-}$ between Hilbert spaces. In passing, we remark that the crosscap partial waves $\Phi_\alpha(\eta)$ are invariant under $\Delta_1 \leftrightarrow \Delta_2$. This is not manifest, but it may be derived from the Jacobi function identity (3.16).

In particular, we can use the alpha space transform (5.13) to decompose the two-point function $\mathcal{G}(\eta)$ as follows:

$$\mathcal{G}(\eta) = \int_{\mathcal{C}} \frac{[d\alpha]}{N_{\text{proj}}(\alpha)}\,\mathcal{G}(\alpha)\Phi_\alpha(\eta) \tag{5.15}$$

where $\mathcal{G}(\alpha)$ represents a spectral density. To recover the CB decomposition (5.15), we notice that

$$\Phi_\alpha(\eta) = \frac{1}{2}\left[Q_{\text{proj}}(\alpha)G_{h+2\alpha}^{\text{proj}}(\eta) + (\alpha \to -\alpha)\right]\,. \tag{5.16}$$

Consequently, $\mathcal{G}(\eta)$ can be written as a sum over all poles $\alpha_k$ circled by $\mathcal{C}$:

$$\mathcal{G}(\eta) = \sum_k c_k G_{h+2\alpha_k}^{\text{proj}}(\eta)\,, \qquad c_k = -\frac{2}{Q_{\text{proj}}(-\alpha_k)}\,\text{Res}\,\mathcal{G}(\alpha)\big|_{\alpha=\alpha_k}\,. \tag{5.17}$$

Once more, the integral (5.15) can be interpreted as an integral over the scalar part of the unitary principal series of the conformal group, cf. our discussion below Eqs. (4.21) and (4.31).

## 5.3 Bootstrap in alpha space

In what follows, we will derive the alpha space version of the bootstrap equation (5.7). We start by inserting the integral decomposition (5.15) into the bootstrap equation (5.7). This yields the following consistency condition:

$$\int \frac{[d\alpha]}{N_{\text{proj}}(\alpha)}\,\mathcal{G}(\alpha)\Phi_\alpha(\eta) = \epsilon\left(\frac{\eta}{1-\eta}\right)^{\Delta_+}\int \frac{[d\alpha]}{N_{\text{proj}}(\beta)}\,\mathcal{G}(\beta)\Phi_\beta(1-\eta) \tag{5.18}$$

for some sign $\epsilon = \pm 1$ (see our discussion below Eq. (5.8)). We proceed as in the boundary case and introduce an integral kernel $C(\alpha, \beta)$ which relates the partial waves appearing on both sides of the above equation:

$$\left(\frac{\eta}{1-\eta}\right)^{\Delta_+} \Phi_\beta(1-\eta) = \int \frac{[d\alpha]}{N_{\text{proj}}(\alpha)} C(\alpha, \beta)\Phi_\alpha(\eta). \tag{5.19}$$

Notice that $C(\alpha, \beta)$ depends on $\Delta_1$, $\Delta_2$ and the dimension $d$. The above kernel allows us to recast Eq. (5.18) as an eigenvalue equation, namely

$$\boxed{(\mathsf{C} \cdot \mathcal{G})(\alpha) = \epsilon\, \mathcal{G}(\alpha)} \tag{5.20}$$

where $\mathsf{C}$ is the following linear operator:

$$(\mathsf{C} \cdot f)(\alpha) := \int \frac{[d\beta]}{N_{\text{proj}}(\beta)} C(\alpha, \beta)f(\beta). \tag{5.21}$$

Eq. (5.20) means that crossing symmetry reduces to finding eigenfunctions of the operator $\mathsf{C}$ with eigenvalue $\epsilon$. For simplicity, we will refer to $C(\alpha, \beta)$ as a *crosscap crossing kernel*, although it arises from a consistency condition that is not immediately related to crossing symmetry.

To proceed, we must compute the kernel $C(\alpha, \beta)$. To do so, one writes $C$ as a position-space integral:

$$C(\alpha, \beta) = \int_0^1 d\eta\, w_{\text{proj}}(\eta) \left(\frac{\eta}{1-\eta}\right)^{\Delta_+} \Phi_\alpha(\eta)\Phi_\beta(1-\eta). \tag{5.22}$$

The evaluation of this integral will be postponed to the next section. For now, we notice that (5.22) implies that $C(\alpha, \beta)$ obeys the duality relation

$$C(\alpha, \beta|\Delta_1, \Delta_2) = C(\beta, \alpha|d - \Delta_2, d - \Delta_1) \tag{5.23}$$

cf. Eqs. (2.20) and (4.42).

# 6 Analyzing the crossing kernels

At this stage we have encountered three different crossing kernels: a boundary-to-bulk and a bulk-to-boundary kernel $E_{\text{bdy}\to\text{bulk}}(\alpha, \nu)$ resp. $E_{\text{bulk}\to\text{bdy}}(\nu, \alpha)$, which appeared in the context of boundary CFTs, and a kernel $C(\alpha, \beta)$ which was relevant for two-point functions on the crosscap $\mathbb{RP}^d$. In this section we will take a closer look at these crossing kernels: we will compute them and compare them to the $d = 1$ crossing kernel from Ref. [1].

## 6.1 Recap: $d = 1$ crossing kernel

To set the stage and fix some notation, let us briefly review the $d = 1$ crossing kernel constructed in Ref. [1]. It arises in the analysis of a four-point function $\langle \phi_1(x_1)\phi_2(x_2)\phi_3(x_3)\phi_4(x_4) \rangle$ of four operators $\phi_i(x_i)$ in a one-dimensional CFT, having $SL(2, \mathbb{R})$ weights $\mathsf{h}_1, \ldots, \mathsf{h}_4$. After stripping off a scale factor, this correlator depends on a single cross ratio $z \in [0, 1]$. Crossing symmetry relates

two different conformal block expansions, commonly referred to as $s$- and $t$-channel decompositions. The $s$-channel decomposition reads[6]

$$\langle \phi_1(x_1)\phi_2(x_2)\phi_3(x_3)\phi_4(x_4)\rangle \ \sim \ \int \frac{[d\alpha]}{\mathcal{N}_{\bar{p},\bar{q}}(\alpha)} \, F_s(\alpha)\Psi_\alpha^s(z)\,, \quad \Psi_\alpha^s(z) = z^{h_1-h_2}\,\vartheta_\alpha^{(\bar{p},\bar{q})}(z)\,. \tag{6.1}$$

In (6.1) we have omitted an $x$-dependent scaling factor which is fixed by conformal symmetry, and we have introduced the shorthand notation

$$\bar{p} = -h_1 + h_2 + h_3 - h_4 \quad \text{and} \quad \bar{q} = -h_1 + h_2 - h_3 + h_4\,. \tag{6.2}$$

At the same time, the $t$-channel decomposition of the correlator $\langle \phi_1\phi_2\phi_3\phi_4\rangle$ reads

$$\langle \phi_1(x_1)\phi_2(x_2)\phi_3(x_3)\phi_4(x_4)\rangle \ \sim \ \int \frac{[d\alpha]}{\mathcal{N}_{\bar{r},\bar{q}}(\alpha)} \, F_t(\alpha)\Psi_\alpha^t(1-z)\,, \quad \Psi_\alpha^t(z) = z^{h_3-h_2}\,\vartheta_\alpha^{(\bar{r},\bar{q})}(z)\,, \tag{6.3}$$

writing

$$\bar{r} = h_1 + h_2 - h_3 - h_4\,. \tag{6.4}$$

Imposing that both decompositions are identical, one obtains the following alpha space bootstrap equation:

$$F_s(\alpha) = (\mathsf{K} \cdot F_t)(\alpha) \tag{6.5}$$

which has been stated in terms of the following integral operator $\mathsf{K}$:

$$(\mathsf{K} \cdot f)(\alpha) = \int \frac{[d\alpha]}{\mathcal{N}_{\bar{r},\bar{q}}(\alpha)} \, K(\alpha,\beta)f(\beta)\,. \tag{6.6}$$

We will refer to the integral kernel $K(\alpha,\beta) = K(\alpha,\beta|h_1,h_2,h_3,h_4)$ appearing in Eq. (6.6) as the $d=1$ or $SL(2,\mathbb{R})$ crossing kernel. It admits the following integral representation:

$$K(\alpha,\beta|h_1,h_2,h_3,h_4) = \int_0^1 dz\, w_{\bar{p},\bar{q}}(z) \left(\frac{z}{1-z}\right)^{\bar{\lambda}} \vartheta_\alpha^{(\bar{p},\bar{q})}(z)\vartheta_\beta^{(\bar{r},\bar{q})}(z)\,, \quad \bar{\lambda} = 2h_2\,. \tag{6.7}$$

This integral can be performed in closed form, yielding a sum of two $_4F_3(1)$ hypergeometric functions. The result is proportional to a Wilson function $\mathsf{W}_\alpha(\beta;a,b,c,d)$ [95]:

$$\mathsf{W}_\alpha(\beta;a,b,c,d) :=$$
$$\frac{\Gamma(d-a)}{\Gamma(a+b)\Gamma(a+c)\Gamma(d\pm\beta)\Gamma(\tilde{d}\pm\alpha)} \, {}_4F_3\left(\begin{matrix} a+\beta,\ a-\beta,\ \tilde{a}+\alpha,\ \tilde{a}-\alpha \\ a+b,\ a+c,\ 1+a-d \end{matrix}\,;1\right) + (a\leftrightarrow d) \tag{6.8}$$

writing

$$\tilde{a} = \tfrac{1}{2}(a+b+c-d)\,, \quad \tilde{d} = \tfrac{1}{2}(-a+b+c+d)\,, \quad \Gamma(x\pm y) = \Gamma(x+y)\Gamma(x-y)\,. \tag{6.9}$$

A closed-form expression for the $d=1$ kernel $K(\alpha,\beta)$ is then given by

$$K(\alpha,\beta|h_1,h_2,h_3,h_4) = \Gamma(1-h_1+h_2+h_3-h_4)\Gamma(1+h_1+h_2-h_3-h_4)$$
$$\times\ \Gamma(h_1+h_2-\tfrac{1}{2}\pm\alpha)\Gamma(\tfrac{3}{2}-h_1-h_4\pm\beta)\mathsf{W}_\alpha(\beta;\mathcal{P}) \tag{6.10}$$

---

[6]Note: our conventions for $\Psi_\alpha^s(z)$ and $\Psi_\alpha^t(z)$ differ slightly from those in Ref. [1]; the kernel $K(\alpha,\beta)$ is identical.

where the parameters $\mathcal{P}$ are chosen as follows:

$$\mathcal{P} = \left\{ \frac{1}{2} + h_1 - h_4, \frac{1}{2} + h_2 - h_3, h_2 + h_3 - \frac{1}{2}, \frac{3}{2} - h_1 - h_4 \right\}. \tag{6.11}$$

In Ref. [95], the Wilson functions $W_\alpha(\beta; a, b, c, d)$ were used to study a four-parameter family of integral transforms, known as the Wilson transform. The properties of this transform can be used to show that $\mathsf{K}$ defines a unitary map between two Hilbert spaces. To make this concrete, let us define a Hilbert space $\mathcal{H}[h_1, h_2, h_3, h_4]$ as the space of all functions $\phi(\alpha) = \phi(-\alpha)$ on the complex plane that are square-normalizable with respect to

$$(\phi, \chi)_{h_1, h_2, h_3, h_4} = \int \frac{[d\alpha]}{\mathcal{M}(\alpha)} \, \overline{\phi(\alpha)} \chi(\alpha) \,,$$

$$\mathcal{M}(\alpha) = \frac{2\Gamma^2(1 + h_1 + h_2 - h_3 - h_4)\Gamma(\pm 2\alpha)\Gamma(h_2 + h_3 - \frac{1}{2} \pm \alpha)}{\Gamma(\frac{1}{2} + h_2 - h_3 \pm \alpha)\Gamma(\frac{1}{2} + h_1 - h_4 \pm \alpha)\Gamma(\frac{3}{2} - h_1 - h_4 \pm \alpha)} \,. \tag{6.12}$$

The functional properties of $\mathsf{K}$ are then summarized as follows:

**Theorem** (1.1 of [1]). $\mathsf{K}$ *is a unitary operator between* $\mathcal{H}[h_1, h_2, h_3, h_4]$ *and* $\mathcal{H}[h_3, h_2, h_1, h_4]$.

Notice that the domain and image of $\mathsf{K}$ are not identical in general, unless $h_1 = h_3$. The above result is a variant of Theorem 4.12 of [95], which applies to the Wilson transform.

A constructive proof of this theorem can be given by introducing an orthogonal basis for the Hilbert space $\mathcal{H}[h_1, h_2, h_3, h_4]$. Such a basis consists of the functions

$$\xi_n(\alpha; h_1, h_2, h_3, h_4) = \Gamma(1 + h_1 + h_2 - h_3 - h_4)\Gamma(h_2 + h_3 - \tfrac{1}{2} \pm \alpha) \, \mathfrak{p}_n(\alpha; \mathcal{P}), \quad n \in \mathbb{N}, \tag{6.13}$$

where $\mathfrak{p}_n(\alpha; a, b, c, d)$ denotes a Wilson polynomial [96–98]:

$$\mathfrak{p}_n(\alpha; a, b, c, d) = (a+b)_n(a+c)_n(a+d)_n \, {}_4F_3\left( \begin{matrix} -n, a + \alpha, a - \alpha, n + a + b + c + d - 1 \\ a + b, a + c, a + d \end{matrix} ; 1 \right). \tag{6.14}$$

The proof relies critically on the following identity

$$\left( \mathsf{K} \cdot \xi_n(\,\cdot\,; h_1, h_2, h_3, h_4) \right)(\alpha) = (-1)^n \xi_n(\alpha; h_3, h_2, h_1, h_4) \tag{6.15}$$

which paraphrases Theorem 6.7 of Ref. [95].

## 6.2 Relating the different kernels

The reader may notice that both the BCFT kernel and the crosscap kernel look similar to the $d = 1$ kernel defined in (6.7). In this subsection we will make the relation between these different kernels precise. It will be easiest to proceed in two steps. First, we introduce a "master" kernel $Z(\alpha, \beta)$ which depends on four parameters:

$$Z(\alpha, \beta | p, q, r, \lambda) := \int_0^1 dx \, w_{p,q}(x) \left( \frac{x}{1-x} \right)^\lambda \vartheta_\alpha^{(p,q)}(x) \vartheta_\beta^{(r,q)}(1-x) \,. \tag{6.16}$$

The parameters $\{p, q, r\}$ encode Jacobi functions, whereas $\lambda$ plays the role of a scaling dimension appearing in a crossing equation. Comparing (6.16) to (6.7), we see that

$$Z(\alpha, \beta | \lambda, p, q, r) = K(\alpha, \beta | \bar{\mathsf{h}}_1, \bar{\mathsf{h}}_2, \bar{\mathsf{h}}_3, \bar{\mathsf{h}}_4) \tag{6.17}$$

where on the RHS we have chosen the weights $\bar{\mathsf{h}}_i$ as follows:

$$\begin{pmatrix} \bar{\mathsf{h}}_1 \\ \bar{\mathsf{h}}_2 \\ \bar{\mathsf{h}}_3 \\ \bar{\mathsf{h}}_4 \end{pmatrix} = \frac{1}{2} \begin{pmatrix} \lambda - p - q \\ \lambda \\ \lambda - q - r \\ \lambda - p - r \end{pmatrix}. \tag{6.18}$$

In other words, we conclude that *any* crossing kernel of the form (6.16) can be expressed as the $SL(2, \mathbb{R})$ kernel $K(\alpha, \beta | \mathsf{h}_1, \mathsf{h}_2, \mathsf{h}_3, \mathsf{h}_4)$, after setting the external weights $\mathsf{h}_1, \mathsf{h}_2, \mathsf{h}_3, \mathsf{h}_4$ to appropriate values. Next, we notice that both the boundary and crosscap kernels are special cases of the master kernel (6.16). Combining these two facts, we arrive at the surprising conclusion that $E_{\mathrm{bdy}\to\mathrm{bulk}}$, $E_{\mathrm{bulk}\to\mathrm{bdy}}$ and $C$ can be obtained as limits of $K(\alpha, \beta)$. In particular, we can use this to find closed-form expressions for the boundary and crosscap kernels without computing any integrals.

Let us spell out the above identification in more detail, starting with the crosscap kernel $C(\alpha, \beta)$. By comparing its integral expression (5.22) to the master kernel (6.16), we see that

$$\begin{aligned} C(\alpha, \beta) &= Z(\alpha, \beta | \Delta_1,\, h - 1,\, 2\Delta_-,\, h - 1) \\ &= K\left(\alpha, \beta | \tfrac{1}{2}(\Delta_2 + 1 - h),\, \tfrac{1}{2}\Delta_1,\, \tfrac{1}{2}(\Delta_2 + 1 - h),\, \tfrac{1}{2}\Delta_1 + 1 - h\right). \end{aligned} \tag{6.19a}$$

Likewise, starting from the integral representation (4.41) of the two BCFT kernels, it follows that

$$\begin{aligned} E_{\mathrm{bdy}\to\mathrm{bulk}}(\alpha, \nu) &= Z(\alpha, \nu | 2\Delta_-,\, h - 1,\, h - 1,\, \Delta_1) \\ &= K\left(\alpha, \nu | \tfrac{1}{2}(\Delta_2 + 1 - h),\, \tfrac{1}{2}\Delta_1,\, \tfrac{1}{2}\Delta_1 + 1 - h,\, \tfrac{1}{2}(\Delta_2 + 1 - h)\right) \end{aligned} \tag{6.19b}$$

and

$$\begin{aligned} E_{\mathrm{bulk}\to\mathrm{bdy}}(\nu, \alpha) &= Z(\nu, \alpha | h - 1,\, h - 1,\, 2\Delta_-,\, \Delta_1) \\ &= K\left(\nu, \alpha | \tfrac{1}{2}\Delta_1 + 1 - h,\, \tfrac{1}{2}\Delta_1,\, \tfrac{1}{2}(\Delta_2 + 1 - h),\, \tfrac{1}{2}(\Delta_2 + 1 - h)\right). \end{aligned} \tag{6.19c}$$

The three different $SL(2, \mathbb{R})$ kernels appearing in (6.19a), (6.19b) and (6.19c) seem to be closely related: they differ only by a permutation of their arguments. We have not found a simple argument to explain this phenomenon.

## 6.3 Crosscap kernel

In this and the following section we will exploit the above identification with the $d = 1$ kernel in more detail. At first we will consider the crosscap case. For concreteness, we can write down a more explicit closed-form formula for the crosscap kernel using (6.10), namely[7]

$$C(\alpha, \beta) = \Gamma^2(h)\Gamma(\Delta_+ - \tfrac{1}{2}h \pm \alpha)\Gamma(\tfrac{3}{2}h - \Delta_+ \pm \beta)\mathsf{W}_\alpha(\beta; \mathcal{P}_{\mathrm{proj}}),$$

$$\mathcal{P}_{\mathrm{proj}} = \left\{ \frac{h}{2} + \Delta_-,\, \frac{h}{2} - \Delta_-,\, \Delta_+ - \frac{h}{2},\, \frac{3h}{2} - \Delta_+ \right\}. \tag{6.20}$$

---

[7]In order to verify Eq. (5.23), it is useful to notice that $\mathsf{W}_\alpha(\beta; \mathcal{P}_{\mathrm{proj}})$ is symmetric under $\alpha \leftrightarrow \beta$.

Moreover, using the results described in Sec. 6.1, we can prove a structural fact about the linear operator $\mathsf{C}$. To do so, we introduce a function space $\mathcal{H}_{\text{proj}}$ which depends on $d$ and the external dimensions $\Delta_1, \Delta_2$. $\mathcal{H}_{\text{proj}}$ consists of all alpha space functions $f(\alpha)$ that are even and $L^2$ with respect to the following inner product:

$$\left(f, g\right)_{\text{proj}} = \int \frac{[d\alpha]}{\mathcal{M}_{\text{proj}}(\alpha)} \, \overline{f(\alpha)} g(\alpha) \,,$$

$$\mathcal{M}_{\text{proj}}(\alpha) = \frac{2\Gamma^2(h)\Gamma(\pm 2\alpha)\Gamma(\Delta_+ - \frac{1}{2}h \pm \alpha)}{\Gamma(\frac{1}{2}h + \Delta_- \pm \alpha)\Gamma(\frac{1}{2}h - \Delta_- \pm \alpha)\Gamma(\frac{3}{2}h - \Delta_+ \pm \alpha)} \,. \quad (6.21)$$

Then we obtain the following theorem:

**Theorem** (1). *The integral operator $\mathsf{C}$ is a unitary map $\mathcal{H}_{\text{proj}} \to \mathcal{H}_{\text{proj}}$ obeying $\mathsf{C}^2 = \text{id}$.*

Being completely explicit, the fact that $\mathsf{C}$ is an isometry means that

$$\left(f, g\right)_{\text{proj}} = \left(\mathsf{C} \cdot f, \mathsf{C} \cdot g\right)_{\text{proj}} \quad (6.22)$$

for any two functions $f, g \in \mathcal{H}_{\text{proj}}$. This generalizes a known fact: for $\Delta_1 = \Delta_2$ and $d = 2$, we recover Theorem 1.2 of [1].

## 6.4  Boundary kernels

Next, let us turn our attention to the two BCFT kernels, $E_{\text{bdy}\to\text{bulk}}$ and $E_{\text{bulk}\to\text{bdy}}$. We will start by providing explicit formulas for these two kernels. Using (6.19), we find

$$E_{\text{bdy}\to\text{bulk}}(\alpha, \nu) = \Gamma(h)\Gamma(1 + 2\Delta_-)\Gamma(\Delta_+ - \tfrac{1}{2}h \pm \alpha)\Gamma(h + \tfrac{1}{2} - \Delta_2 \pm \nu)\mathsf{W}_\alpha(\nu; \mathcal{P}_{\text{bdy}}) \,,$$

$$\mathcal{P}_{\text{bdy}} = \left\{\frac{1}{2}, \, h - \frac{1}{2}, \, \Delta_1 + \frac{1}{2} - h, \, \frac{1}{2} + h - \Delta_2\right\} \,. \quad (6.23)$$

and likewise[8]

$$E_{\text{bulk}\to\text{bdy}}(\nu, \alpha) = \Gamma(h)\Gamma(1 + 2\Delta_-)\Gamma(\Delta_1 + \tfrac{1}{2} - h \pm \nu)\Gamma(\tfrac{3}{2}h - \Delta_+ \pm \alpha)\mathsf{W}_\nu(\alpha; \mathcal{P}_{\text{bulk}}) \,,$$

$$\mathcal{P}_{\text{bulk}} = \left\{\Delta_+ - \frac{h}{2}, \, \frac{3h}{2} - \Delta_+, \, 1 - \frac{h}{2} + \Delta_-, \, \frac{h}{2} + \Delta_-\right\} \,. \quad (6.24)$$

Our aim is to prove a structural theorem about the operators $E_{\text{bdy}\to\text{bulk}}$ and $E_{\text{bulk}\to\text{bdy}}$, similar to the one proved above for $\mathsf{C}$. We proceed by defining a Hilbert space $\mathcal{H}_{\text{bdy}}$ on the boundary side, containing all functions $f(\nu) = f(-\nu)$ that are square normalizable with respect to

$$\left(f, g\right)_{\text{bdy}} = \int \frac{[d\nu]}{\mathcal{M}_{\text{bdy}}(\nu)} \, \overline{f(\nu)} g(\nu) \,,$$

$$\mathcal{M}_{\text{bdy}}(\nu) = \frac{2\Gamma^2(h)\Gamma(\pm 2\nu)\Gamma(\Delta_1 + \frac{1}{2} - h \pm \nu)}{\Gamma(\frac{1}{2} \pm \nu)\Gamma(h - \frac{1}{2} \pm \nu)\Gamma(\frac{1}{2} + h - \Delta_2 \pm \nu)} \,. \quad (6.25)$$

---

[8]Using a "duality" property of the Wilson function [95], it can be shown that $\mathsf{W}_\alpha(\nu; \mathcal{P}_{\text{bdy}}) = \mathsf{W}_\nu(\alpha; \mathcal{P}_{\text{bulk}})$. In turn, this implies Eq. (4.42).

Likewise, bulk alpha space functions naturally live in $\mathcal{H}_{\text{bulk}}$, defined by the inner product

$$\left(f, g\right)_{\text{bulk}} = \int \frac{[d\alpha]}{\mathcal{M}_{\text{bulk}}(\alpha)} \, \overline{f(\alpha)} g(\alpha) \,,$$

$$\mathcal{M}_{\text{bulk}}(\alpha) = \frac{2\Gamma^2(1 + 2\Delta_-)\Gamma(\pm 2\alpha)\Gamma(\Delta_+ - \frac{1}{2}h \pm \alpha)}{\Gamma(\frac{1}{2}h + \Delta_- \pm \alpha)\Gamma(\frac{3}{2}h - \Delta_+ \pm \alpha)\Gamma(\Delta_- + 1 - \frac{1}{2}h \pm \alpha)} \,. \quad (6.26)$$

Then the analog of the previous result reads:

**Theorem** (2). *The operator* $\mathsf{E}_{\text{bdy}\to\text{bulk}} : \mathcal{H}_{\text{bdy}} \to \mathcal{H}_{\text{bulk}}$ *is unitary, and* $\mathsf{E}_{\text{bulk}\to\text{bdy}} : \mathcal{H}_{\text{bulk}} \to \mathcal{H}_{\text{bdy}}$ *is its inverse, i.e.*

$$\mathsf{E}_{\text{bdy}\to\text{bulk}} \cdot \mathsf{E}_{\text{bulk}\to\text{bdy}} \;=\; \mathsf{E}_{\text{bulk}\to\text{bdy}} \cdot \mathsf{E}_{\text{bdy}\to\text{bulk}} \;=\; \text{id} \,. \quad (6.27)$$

The fact that these operators are norm-preserving means that for any two functions $f, g \in \mathcal{H}_{\text{bdy}}$ we have

$$\left(f, g\right)_{\text{bdy}} = \left(\mathsf{E}_{\text{bdy}\to\text{bulk}} \cdot f, \, \mathsf{E}_{\text{bdy}\to\text{bulk}} \cdot g\right)_{\text{bulk}} , \quad (6.28)$$

and likewise

$$\left(f, g\right)_{\text{bulk}} = \left(\mathsf{E}_{\text{bulk}\to\text{bdy}} \cdot f, \, \mathsf{E}_{\text{bulk}\to\text{bdy}} \cdot g\right)_{\text{bdy}} \quad (6.29)$$

provided that $f, g \in \mathcal{H}_{\text{bulk}}$.

## 6.5  Mean-field solutions

One advantage of the identification of the crossing kernels with Wilson functions is that they allow us to find infinitely many solutions to crossing symmetry directly in alpha space. For $d = 1$ CFTs this was shown in [1]. Here we will describe a similar result for boundary and crosscap CFTs.

Let us start by considering crosscap CFTs. First, notice that a basis for the Hilbert space $\mathcal{H}_{\text{proj}}$ is given by the functions

$$\xi_n^{\text{proj}}(\alpha) := \Gamma(h)\Gamma(\Delta_+ - \tfrac{1}{2}h \pm \alpha) \, \mathfrak{p}_n(\alpha; \mathcal{P}_{\text{proj}}) \quad (6.30)$$

which are proportional to Wilson polynomials. We claim that these basis functions $\xi_n^{\text{proj}}(\alpha)$ transform in a simple way under crossing. This is a consequence of (6.15): by setting $\mathsf{h}_1, \dots, \mathsf{h}_4$ to appropriate values, it can be shown that the basis functions satisfy

$$(\mathsf{C} \cdot \xi_n^{\text{proj}})(\alpha) = (-1)^n \xi_n^{\text{proj}}(\alpha) \,. \quad (6.31)$$

Comparing this to the crosscap bootstrap equation (5.20), it follows that any sum of the form

$$\mathcal{G}(\alpha) = \sum_{n \text{ even/odd}} c_n \xi_n^{\text{proj}}(\alpha) \quad (6.32)$$

is a consistent solution with $\epsilon = 1$ (even $n$) resp. $\epsilon = -1$ (odd $n$).

We remark that the functions $\xi_n^{\text{proj}}$ behave as mean-field solutions to crossing, in the sense that they have an integer-spaced spectrum. To see this, notice from (6.30) that $\xi_n^{\text{proj}}(\alpha)$ has poles at

$\alpha = \Delta_+ - h/2 + \mathbb{N}$. As a consequence of (5.17), the spectrum of $\xi_n^{\text{proj}}(\alpha)$ consists of an infinite tower of primary states with dimensions $\Delta_1 + \Delta_2 + 2\mathbb{N}$. In other words, $\xi_n^{\text{proj}}$ admits a CB decomposition of the following form:

$$\xi_n^{\text{proj}}(\eta) = \sum_{m=0}^{\infty} k_{m,n} \, G_{\Delta_1 + \Delta_2 + 2m}^{\text{proj}}(\eta) \tag{6.33}$$

for some coefficients $k_{m,n}$ that are easily computable. In particular, it follows that $\xi_n^{\text{proj}}$ does not have a unit operator contribution.

It is straightforward to compute the position-space behaviour of the above basis functions. The result is

$$\xi_n^{\text{proj}}(\eta) = \frac{n!(h)_n \Gamma(\Delta_1 + n)\Gamma(\Delta_2 + n)}{(2h-1)_n} \, \eta^{\Delta_+} \, \mathcal{C}_n^{(h-1/2)}(1 - 2\eta) \,, \tag{6.34}$$

where the functions $\mathcal{C}_n^{(\lambda)}$ are Gegenbauer polynomials — see Appendix A for details of this computation. For certain integer values of $d$, these are Chebyshev polynomials of the first ($d = 1$) or second ($d = 3$) kind, or Legendre polynomials ($d = 2$). Using the fact that the polynomial $\mathcal{C}_n$ has parity $(-1)^n$, it follows that under crossing the solutions $\xi_n^{\text{proj}}(\eta)$ transform as

$$\xi_n^{\text{proj}}(\eta) = (-1)^n \left( \frac{\eta}{1 - \eta} \right)^{\Delta_+} \xi_n^{\text{proj}}(1 - \eta) \,. \tag{6.35}$$

This is an alternative way to derive Eq. (6.31).

A similar class of mean-field solutions exists in the boundary case. There, we remark that a basis for the Hilbert spaces $\mathcal{H}_{\text{bdy}}$ resp. $\mathcal{H}_{\text{bulk}}$ is given by

$$\xi_n^{\text{bdy}}(\nu) = \Gamma(h)\Gamma(\Delta_1 + \tfrac{1}{2} - h \pm \nu) \, \mathfrak{p}_n(\nu; \mathcal{P}_{\text{bdy}}) \,, \tag{6.36a}$$

$$\xi_n^{\text{bulk}}(\alpha) = \Gamma(1 + 2\Delta_-)\Gamma(\Delta_+ - \tfrac{1}{2}h \pm \alpha) \, \mathfrak{p}_n(\alpha; \mathcal{P}_{\text{bulk}}) \,. \tag{6.36b}$$

Under crossing, these basis functions transform as

$$\begin{pmatrix} \mathsf{E}_{\text{bdy} \to \text{bulk}} & 0 \\ 0 & \mathsf{E}_{\text{bulk} \to \text{bdy}} \end{pmatrix} \cdot \begin{pmatrix} \xi_n^{\text{bdy}} \\ \xi_n^{\text{bulk}} \end{pmatrix} = (-1)^n \begin{pmatrix} \xi_n^{\text{bulk}} \\ \xi_n^{\text{bdy}} \end{pmatrix} \tag{6.37}$$

as follows from (6.15). Using (6.37) we can thus easily generate solutions to the boundary bootstrap equation (4.40):

$$\mathcal{F}_{\text{bdy}}(\nu) = \sum_{n \text{ even}} c_n \xi_n^{\text{bdy}}(\nu) \,, \quad \mathcal{F}_{\text{bulk}}(\alpha) = \sum_{n \text{ even}} c_n \xi_n^{\text{bulk}}(\alpha) \,, \tag{6.38}$$

where the coefficients $c_n$ are arbitrary.

The basis functions from (6.36) once more have a mean-field type spectrum. For instance, $\xi_n^{\text{bulk}}(\alpha)$ has poles at $\alpha = \Delta_+ - \tfrac{1}{2}h + \mathbb{N}$, hence its CB decomposition is a sum over conformal blocks with dimensions $\Delta_1 + \Delta_2 + 2\mathbb{N}$. Likewise, the CB decomposition of $\xi_n^{\text{bdy}}(\nu)$ runs over boundary states with dimensions $\Delta_1 + \mathbb{N}$.[9]

For completeness, we compute that in position space the solutions (6.36) read

$$\begin{bmatrix} \xi_n^{\text{bulk}}(\rho) \\ \xi_n^{\text{bdy}}(\rho) \end{bmatrix} = n!\Gamma(\Delta_1 + n)\Gamma(\Delta_1 + 1 - h + n) \begin{bmatrix} \rho^{\Delta_+}(1 - \rho)^{\Delta_-} \\ (-1)^n (1 - \rho)^{\Delta_1} \end{bmatrix} P_n^{(h-1, 2\Delta_-)}(1 - 2\rho) \tag{6.39}$$

---

[9] Notice that the symmetry $\Delta_1 \leftrightarrow \Delta_2$ is manifestly broken, cf. our remarks around Eq. (4.22).

as is shown in Appendix A. These bulk and boundary basis functions are related as follows

$$\left(\frac{1-\rho}{\rho}\right)^{\Delta_+}\xi_n^{\text{bulk}}(\rho) = (-1)^n \xi_n^{\text{bdy}}(\rho)\,, \tag{6.40}$$

providing an alternative proof of Eq. (6.37).

# 7 Discussion

In this paper we investigated CFT consistency conditions through the lens of Sturm-Liouville theory, specializing to boundary and crosscap CFTs in arbitrary spacetime dimension $d$. This led to a generalization of the alpha space transform developed in Ref. [1]. In particular, we found that in the boundary and crosscap cases this transform is closely related to the Jacobi transform, as had been established for $d = 1$ CFTs in previous work.

A key result of this paper was the computation of the relevant crossing kernels. We showed that, surprisingly, the boundary and crosscap kernels can both be interpreted as limits of the $d = 1$ or $SL(2,\mathbb{R})$ kernel constructed in [1]. For bootstrap purposes, this has a direct implication: any method to solve $d = 1$ bootstrap equations can directly be used to solve boundary and crosscap bootstrap equations as well. In hindsight, this fact could have been derived by inspecting the relevant position-space bootstrap equations. However, the simplicity of our derivation shows that some aspects of crossing symmetry are perhaps more transparent in alpha space.

In our exposition, the above relation between different kernels appeared as a fortunate co-incidence. At present we are not aware of an obvious group-theoretical reason why such an identification should exist. In future work, it would certainly be interesting to investigate this starting from conformal representation theory.

From the mathematical point of view, we found that the boundary and crosscap bootstrap equations in alpha space were closely related to the Wilson transform introduced in Ref. [95]. This allowed us to construct infinitely many solutions to crossing in alpha space. In this work we did not attempt to extract interesting bootstrap constraints from the alpha space formalism (although some ideas in that direction were sketched in [1]). An ambitious goal for the future would be to construct new, analytic solutions to crossing symmetry. We believe that the $d = 2$ Liouville CFT literature could provide a starting point for such an investigation. The reason is that solutions to Liouville theory in the presence of a boundary and on the crosscap are understood analytically, see e.g. [99–104].[10]

There are several possible directions in which this work can be generalized. In the introduction, we already mentioned the more general problem of constructing the crossing kernel for four-point functions in $d$-dimensional CFTs. Some non-trivial features of that kernel will already appear at the level of boundary and crosscap CFTs. For instance, the alpha space analysis of conserved current or stress tensor two-point functions will likely involve some new mathematical machinery, unrelated to the Jacobi and Wilson transforms used in this paper. Likewise, it should be possible to examine codimension-$p$ or supersymmetric defects in alpha space.

Finally, one may ask to which extent the formalism in this work admits a holographic interpretation. For one, it should be possible to compare our alpha space formalism to the more

---

[10]See also Refs. [105–110] for similar results in the supersymmetric case.

familiar Mellin space language. The dictionary between BCFT or crosscap correlators and Mellin amplitudes does not appear to have been worked out in the literature yet, hence this question should be addressed first. We also point to the recent work [111] which describes an alternative decomposition of BCFT correlators in terms of AdS geodesic operators. Finally, the $d$-dimensional crosscap kernel has been mentioned in relation to AdS bulk locality in Ref. [80]. It is certainly interesting to investigate whether some questions in that context can be addressed using alpha space methods.

### Acknowledgments

We are grateful to Liam Fitzpatrick, Hugh Osborn, Leonardo Rastelli and Balt van Rees for comments and/or discussions. This research was supported in part by Perimeter Institute for Theoretical Physics. Research at Perimeter Institute is supported by the Government of Canada through Industry Canada and by the Province of Ontario through the Ministry of Research and Innovation.

## A    Examples of alpha space transforms

For reference, we compute the alpha space transform of two simple classes of position-space functions: power laws and Jacobi polynomials.

### A.1    Power laws

Let us consider the alpha space transform for the following double power law in position space:

$$x \; \mapsto \; \frac{x^{\ell_1}}{(1-x)^{\ell_2}} \, . \tag{A.1}$$

We will be interested in the alpha space transform of (A.1) for both the bulk/boundary and crosscap transforms. The results are variants of the following integral:[11]

$$\mathcal{Y}_{p,q}(\alpha; \ell_1, \ell_2) := \int_0^1 dx \, w_{p,q}(x) \, \frac{x^{\ell_1}}{(1-x)^{\ell_2}} \, \vartheta_\alpha^{(p,q)}(x) \tag{A.2}$$

$$= \frac{\Gamma(1+p)\Gamma\big(\ell_1 - \tfrac{1}{2}(1+p+q) \pm \alpha\big)}{\Gamma(\ell_1)\Gamma(\ell_1 - q)}$$

$$\times \; {}_3F_2\left( \begin{matrix} \ell_1 - \tfrac{1}{2}(1+p+q) + \alpha, \; \ell_1 - \tfrac{1}{2}(1+p+q) - \alpha, \; \ell_2 \\ \ell_1, \; \ell_1 - q \end{matrix} \; ; 1 \right) \tag{A.3}$$

$$= \mathcal{Y}_{p,-q}(\alpha; \ell_1 - q, \ell_2).$$

---

[11]The formula (A.3) converges only if $p + 1 > \ell_2$, but it can be analytically continued to generic values of $\ell_1, \ell_2$.

Specializing to the three different alpha space transforms in this paper, we find that

$$\int_0^1 d\rho\, w_{\text{bulk}}(\rho)\, \frac{\rho^{\ell_1}}{(1-\rho)^{\ell_2}}\, \Psi_\alpha^{\text{bulk}}(\rho) = \mathcal{Y}_{2\Delta_-, h-1}(\alpha; \ell_1 + \Delta_-, \ell_2 + \Delta_-)\,, \tag{A.4a}$$

$$\int_0^1 d\rho\, w_{\text{bdy}}(\rho)\, \frac{\rho^{\ell_1}}{(1-\rho)^{\ell_2}}\, \Psi_\nu^{\text{bdy}}(\rho) = \mathcal{Y}_{h-1, h-1}(\nu; -\ell_2, -\ell_1)\,, \tag{A.4b}$$

$$\int_0^1 d\eta\, w_{\text{proj}}(\eta)\, \frac{\eta^{\ell_1}}{(1-\eta)^{\ell_2}}\, \Phi_\alpha(\eta) = \mathcal{Y}_{h-1, 2\Delta_-}(\alpha; \ell_1 + \Delta_-, \ell_2)\,. \tag{A.4c}$$

As a consistency check of Eq. (A.4), we can try to recover the position space function (A.1) from the alpha space densities (A.4) by taking residues. The results are conveniently expressed in terms of the following coefficients:

$$Y_n^{p,q}(\ell_1, \ell_2) := -\text{Res}\, \frac{2}{Q_{p,q}(-\alpha)}\, \mathcal{Y}_{p,q}(\alpha; \ell_1, \ell_2)\big|_{\alpha = \ell_1 - \frac{1}{2}(1+p+q)+n} \tag{A.5}$$

$$= \frac{(-1)^n}{n!}\, \frac{(\ell_1)_n (\ell_1 - q)_n}{(2\ell_1 - 1 - p - q + n)_n}\, {}_3F_2\left( \begin{matrix} -n,\, 2\ell_1 - 1 - p - q + n,\, \ell_2 \\ \ell_1,\, \ell_1 - q \end{matrix} ; 1 \right)\,. \tag{A.6}$$

For the different alpha space transforms, we find

$$\frac{\rho^{\ell_1}}{(1-\rho)^{\ell_2}} = \sum_{n=0}^{\infty} Y_n^{2\Delta_-, h-1}(\ell_1 + \Delta_-, \ell_2 + \Delta_-) G_{2\ell_1 + 2n}^{\text{bulk}}(\rho) \tag{A.7a}$$

$$= \sum_{n=0}^{\infty} Y_n^{h-1, h-1}(-\ell_2, -\ell_1) G_{-\ell_2 + n}^{\text{bdy}}(\rho) \tag{A.7b}$$

$$\frac{\eta^{\ell_1}}{(1-\eta)^{\ell_2}} = \sum_{n=0}^{\infty} Y_n^{h-1, 2\Delta_-}(\ell_1 + \Delta_-, \ell_2) G_{2\ell_1 + 2n}^{\text{proj}}(\eta)\,. \tag{A.7c}$$

Notice that the above coefficients are all invariant under $\Delta_1 \leftrightarrow \Delta_2$, as follows from

$$Y_n^{p,q}(\ell_1, \ell_2) = Y_n^{-p,q}(\ell_1 - p, \ell_2 - p) = Y_n^{p,-q}(\ell_1 - q, \ell_2)\,. \tag{A.8}$$

Eqs. (A.7a), (A.7b), (A.7c) can be easily checked by Taylor expanding around $\rho = 0, 1$ and $\eta = 0$. Also, notice that for generic values of $\ell_1, \ell_2$ the CB decompositions (A.7) do not have positivity properties. The coefficients $Y_n$ only become sign-definite in special limits, for instance

$$Y_n^{p,q}(\ell_1, \ell_1) = \frac{(\ell_1)_n (\ell_1 - p)_n}{n!(2\ell_1 - 1 - p - q + n)_n}\,, \quad Y_n^{p,q}(\ell_1, \ell_1 - q) = \frac{(\ell_1 - q)_n (\ell_1 - p - q)_n}{n!(2\ell_1 - 1 - p - q + n)_n}\,. \tag{A.9}$$

## A.2 Mapping Jacobi polynomials to Wilson polynomials

Second, we consider the alpha space transform for a special class of rational functions in position space. We use the well-known fact that the Jacobi transform maps Jacobi polynomials to Wilson polynomials, as was first noted in [112]. In our conventions, this relation reads

$$\int_0^1 dx\, w_{p,q}(x)\, x^{\frac{1}{2}(p+q+r+s)+1} P_n^{(r,p)}(1 - 2x) \vartheta_\alpha^{(p,q)}(x)$$

$$= \frac{\Gamma(1+p)\Gamma\left(\frac{1}{2}(r+s+1) \pm \alpha\right)}{n!\Gamma\left(\frac{1}{2}(p+q+r+s)+1+n\right)\Gamma\left(\frac{1}{2}(p-q+r+s)+1+n\right)}$$

$$\times\, \mathfrak{p}_n\left(\alpha; \tfrac{1}{2}(p+q+1), \tfrac{1}{2}(p-q+1), \tfrac{1}{2}(r+s+1), \tfrac{1}{2}(r-s+1)\right)\,. \tag{A.10}$$

For the two alpha space transforms associated with BCFTs, we obtain:

$$\int_0^1 d\rho\, w_{\text{bulk}}(\rho)\rho^{\Delta_+}(1-\rho)^{\Delta_-}P_n^{(h-1,2\Delta_-)}(1-2\rho)\Psi_\alpha^{\text{bulk}}(\rho) = \frac{\xi_n^{\text{bulk}}(\alpha)}{n!\Gamma(\Delta_1+n)\Gamma(\Delta_1+1-h+n)}\,, \quad \text{(A.11a)}$$

$$\int_0^1 d\rho\, w_{\text{bdy}}(\rho)(1-\rho)^{\Delta_1}P_n^{(h-1,2\Delta_-)}(1-2\rho)\Psi_\nu^{\text{bdy}}(\rho) = \frac{(-1)^n\xi_n^{\text{bdy}}(\nu)}{n!\Gamma(\Delta_1+n)\Gamma(\Delta_1+1-h+n)}\,. \quad \text{(A.11b)}$$

The alpha space functions appearing on the RHS are defined in Eq. (6.36). These identities are derived from (A.10) with $(p,q,r,s) = (2\Delta_-, h-1, h-1, 2\Delta_+-2h)$ resp. $(h-1, h-1, 2\Delta_-, 2\Delta_+-2h)$. Likewise, specializing to the parameters $(p,q,r,s) = (h-1, 2\Delta_-, h-1, 2\Delta_+-h)$ we find

$$\int_0^1 d\eta\, w_{\text{proj}}(\eta)\,\eta^{\Delta_+}\mathcal{C}_n^{(h-1/2)}(1-2\eta)\Phi_\alpha(\eta) = \frac{(2h-1)_n}{n!(h)_n\Gamma(\Delta_1+n)\Gamma(\Delta_2+n)}\,\xi_n^{\text{proj}}(\alpha) \qquad \text{(A.12)}$$

where the functions $\xi_n^{\text{proj}}(\alpha)$ were defined in Eq. (6.30) and $\mathcal{C}_n^{(\lambda)}(x)$ denotes a Gegenbauer polynomial:

$$\mathcal{C}_n^{(\lambda)}(x) = \frac{(2\lambda)_n}{(\lambda+\frac{1}{2})_n}P_n^{(\lambda-1/2,\lambda-1/2)}(x)\,. \qquad \text{(A.13)}$$

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
