# Peer review of "Crossing Kernels for Boundary and Crosscap CFTs"

_SciPost Physics_

## Round 2 · Referee Report · Anonymous (Referee 1) · 2017-11-29

Strengths

  1. Rigorous in all areas that a physicist would expect.
  2. Full of citations to the mathematical literature.
  3. Self-contained with a nice pedagogical section at the beginning.
  4. Extremely relevant to current problems.
  5. Well organized and a pleasure to read.

Weaknesses

  1. Some context about the crossing kernel appears to be missing.
  2. Most of the work was done in the author's previous paper.

Report

This is the second major study of alpha space --- the integral transform that projects a correlator onto a normalizable eigenfunction of the conformal Casimir. The paper, by one of the two original authors, shows that for a boundary or crosscap CFT in arbitrary dimension, the scalar operator crossing kernel is a special case of the crossing kernel for 1D CFTs in flat space. The result is a unified framework for all of the main bootstrap problems that involve a single cross-ratio. The detail and clarity also help make this an exciting paper that SciPost should publisher.

Despite the extensive bibliography, there is one place where I would include a new paper (1705.05362) and one place where I would move existing references so that they appear earlier.

Requested changes

  1. Check the references to equation (2.12). I think they should say (2.10) instead.
  2. Similarly, check the first three references to equation (5.15). I think they should say (5.5).
  3. It is redundant to say "boundary BCFT" in the first sentence of section 4.
  4. The last paragraph, saying that BCFT Mellin amplitudes are unexplored, was perfectly correct when the paper was written. However, it would now be appropriate to update it so that it cites Rastelli and Zhou.
  5. Finally, I would consider rewording the sentence with footnote 1. Reference 47 is far from the only paper that mentions the crossing kernel and studies of it in Liouville theory have led to some celebrated results. I agree that the alpha space papers are the first ones to find model-independent expressions, but calling the object itself "new" is probably overstating things.

---

## Round 2 · Referee Report · Anonymous (Referee 2) · 2017-12-19

Strengths

1 Very precise and carefully written. Although technical not too difficult to read.
2 Very topical area

Weaknesses

1 Result are mostly a precursor to other work

Report

Although rather technical and more in the area of mathematical rather than theoretical physics this is a nice paper
which merits publication.
One issue which might be mentioned if only in passing is whether the crossing kernels are idempotent as I imagine they should be. Perhaps this is obvious or the result of some basic theorem. More mathematically do the crossing kernels define compact operators on the relevant space. This would be relevant in applications.
There are a few typos which should be fixed and which I list below.

Requested changes

1 just before 2.11 I imagine it should say derived from 2.10 rather than 2.12.
2 In 4.5 the definition should be for Delta_{+-} not just Delta_+
3 The use of the phrase gauging away just before 4.15 is inappropriate, just factoring off would be better.
4 Some references could be updated to include the journal, e.g ref 1 but there are others

---

## Round 2 · Referee Report · Anonymous (Referee 3) · 2017-12-30

Strengths

1-very clearly written
2-several examples are provided
3-mathematically rigorous
4-method applicable in several situations

Weaknesses

1- the notation in some parts of the text is a bit heavy 2-reference to previous work is not always well explained, for non expert readers can be a bit hard to follow all the logical steps

Report

I think the paper contains interesting results and can be very useful in the study of a hot topic nowadays as CFTs with boundaries/interfaces. The paper is clearly written and has a good mathematical rigour which strengthen the results.

Requested changes

1- (4.5), is it meant to be \Delta{\pm}? if not please explain how to interpret the signs.

---

## Editorial Decision

awaiting_resubmission